# Social Motion Prediction with Cognitive Hierarchies

**Wentao Zhu** [1*]    **Jason Qin** [1*]    **Yuke Lou** [1]    **Hang Ye** [1]
**Xiaoxuan Ma** [1]    **Hai Ci** [1]    **Yizhou Wang** [1,2]

[1] Center on Frontiers of Computing Studies, School of Computer Science, Peking University
[2] Institute for Artificial Intelligence, Peking University

## Abstract

Humans exhibit a remarkable capacity for anticipating the actions of others and planning their own actions accordingly. In this study, we strive to replicate this ability by addressing the social motion prediction problem. We introduce a new benchmark, a novel formulation, and a cognition-inspired framework. We present `Wusi`, a 3D multi-person motion dataset under the context of team sports, which features intense and strategic human interactions and diverse pose distributions. By reformulating the problem from a multi-agent reinforcement learning perspective, we incorporate behavioral cloning and generative adversarial imitation learning to boost learning efficiency and generalization. Furthermore, we take into account the cognitive aspects of the human social action planning process and develop a cognitive hierarchy framework to predict strategic human social interactions. We conduct comprehensive experiments to validate the effectiveness of our proposed dataset and approach.

## 1   Introduction

Human beings are inherently social creatures. Concretely, individuals unconsciously anticipate the actions of others and make informed decisions about their own behaviors in social contexts [13, 19, 54], which enables individuals to cooperate and compete with others in a variety of scenarios, from pedestrian traffic [25, 44] to team sports [64, 50]. Notably, task experts demonstrate exceptional skill in predicting others' movements in advance [37, 70].

To better understand and replicate this ability, the research community has proposed the task of future prediction for multiple interacting agents given their historical observations. The majority of prior work in this area focuses on modeling and predicting agent interactions at the trajectory level [42, 39, 52, 5, 34, 58, 75, 59, 43, 72] and has demonstrated promising results in applications such as autonomous driving [38, 49, 17, 36, 16, 27]. However, trajectory-based approaches can only reflect coarse-grained interactions (*e.g.*, collision avoidance, social distancing) and fail to capture the rich and fine-grained human actions. To this end, some studies have investigated the *multi-person motion prediction* problem, which aims to forecast both trajectories and poses for a group of people [2, 3, 67, 68, 22, 63, 48, 73]. Despite recent advancements, there remain several critical challenges in this field. Firstly, existing multi-person motion datasets are primarily designed for human pose estimation tasks [65, 1, 31], and consequently, do not place particular emphasis on human interactions. Individuals in these datasets tend to move casually and interact with others at random, making future predictions both difficult and less meaningful. Secondly, the majority of prior methods concentrate on developing neural network architectures for end-to-end supervised training while overlooking the cognitive aspects of human social action planning. These two challenges are closely related and necessitate a comprehensive solution.

---

*Equal Contribution.

37th Conference on Neural Information Processing Systems (NeurIPS 2023).

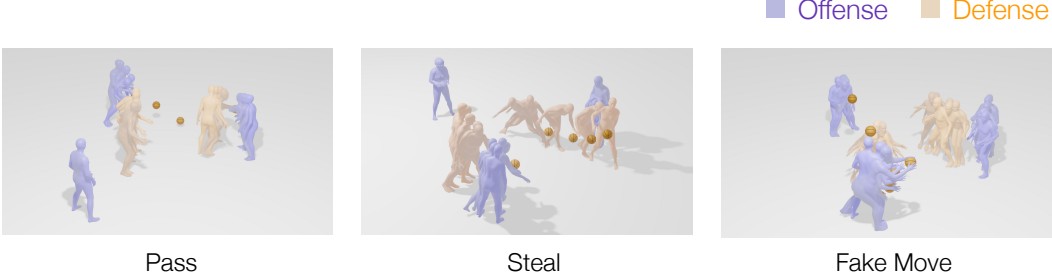

Figure 1: Example sequences from **Wusi** dataset. Three players in purple form the offensive team, and two players in orange form the defensive team. Left: a successful pass for the offensive team. Middle: a successful steal for the defensive team. Right: a successful pass following a fake pass.

In this work, we propose novel perspectives on this problem to address the aforementioned limitations. We begin by constructing a large-scale multi-person 3D motion dataset featuring intense and strategic interactions among participants. To achieve this, we turn to team sports, which offer several inherent advantages: 1) Well-defined game rules and global rewards implicitly constrain and guide individual actions. 2) Participants develop intricate interaction strategies based on their roles, such as planning cooperative actions with teammates while acting adversarially towards opponents. Skilled players even employ sophisticated techniques like deception, *etc*. 3) Human motions exhibit greater dynamism in terms of pose diversity and motion intensity, making motion prediction more challenging than in previous datasets.

Additionally, we present a new formulation of the multi-person motion prediction task as a multi-agent reinforcement learning (MARL) problem. Specifically, we model the task using imitation learning, where the objective is to learn a policy from expert demonstrations. We employ behavioral cloning (BC) [7, 51] to imitate expert behaviors from the dataset. To improve learning efficiency and generalization, we utilize generative adversarial imitation learning (GAIL) [26], aiming to render agents' policy indistinguishable from experts' policy. Furthermore, we propose a framework for human social interactions based on the cognitive hierarchy theory [10]. In particular, we assume that people base their decisions on their predictions regarding the likely actions of others, while others engage in similar decision-making processes from their perspectives. By considering the reasoning steps recursively, we posit that a level $k$ agent takes actions based on the level $k - 1$ agent actions, where $k$ represents the depth of strategic thought. Building on this insight, we develop a computation model that can be elegantly integrated with the MARL formulation.

We summarize our contributions as follows: 1) We present **Wusi**, the first large-scale multi-human 3D motion dataset featuring intense and strategic interactions. We demonstrate that our dataset has greater motion diversity than existing datasets and poses a more significant challenge for the social motion prediction problem. 2) We propose a novel MARL formulation for the problem and develop an imitation learning baseline that combines behavioral cloning and generative adversarial imitation learning. 3) We introduce a cognitive hierarchy framework to model the strategic and game-theoretic human social interactions. Our approach outperforms the state-of-the-art methods in challenging long-term social motion predictions.

## 2 Related work

### 2.1 Multi-agent trajectory prediction

The multi-agent trajectory prediction problem has been extensively studied, especially for interacting traffic participants, *e.g*., pedestrians and vehicles. There exist multiple industrial benchmarks including Argoverse [12], nuScenes [8], Waymo Open Motion Dataset [18]. Meanwhile, various generation methods have been explored to model the interaction among different agents and predict their future trajectories [42, 39, 52, 5, 34, 58, 75, 59, 43, 72]. In this work, we aim to predict intense and strategic interactions for a group of humans with fine-grained body motions.

## 2.2 Multi-person motion dataset

The existing multi-human motion datasets can be categorized into 2D and 3D. The PoseTrack dataset [6, 29] offers video sequences with manually annotated 2D keypoints. However, 2D keypoints fail to accurately represent the proximity of individuals in the real world, and manual annotation is difficult to scale. Meanwhile, 3D multi-person motion datasets can be effectively constructed with motion capture (mocap) systems, including marker-based solutions and markerless ones. 3DPW [65] uses the inertial measurement units (IMUs) to obtain high-quality motion reconstructions. UMPM [61] employs a set of reflective markers to identify the joint positions. However, wearable sensors may struggle to capture complex movements and become expensive as the number of individuals increases. Other datasets, *e.g.*, CMU-Mocap [1], Panoptic [31], MuPoTs-3D [41], ExPI [22], employ markerless mocap systems with multi-view cameras to address the occlusion problem and obtain 3D human motion through triangulation. In this work, we construct a multi-person motion dataset featuring high motion diversity and intense social interactions using multi-view cameras. We provide a comparison with previous datasets in Section 3.3.

## 2.3 Multi-person motion prediction

Joo *et al*. [32] propose to predict human motion conditioned on other individuals' kinesic signals in a triadic haggling scenario. Adeli *et al*. [2] introduce the 2D social motion forecasting (SoMoF) benchmark, which aims to predict multi-person trajectory and pose on the PoseTrack dataset [6]. They also propose a baseline model using a shared GRU encoder and a pooling layer to incorporate social clues. TRiPOD [3] employs attention graphs to characterize the spatiotemporal social interactions. It also considers the joint occlusion and body invisibility issues arising from 2D observations. Futuremotion [67] provides a simple baseline using no social context, yet achieving competitive performance on SoMoF. As 2D multi-person motion prediction faces challenges such as data scarcity, depth ambiguity, and occlusions, more recent research has shifted towards exploring the multi-person motion prediction problem in 3D. Wang *et al*. [68] propose a multi-range Transformer to separately encode the local and global motion history. Guo *et al*. [22] design a cross-interaction-attention (XIA) module to model close interactions between pairs in duo dance scenarios. Vendrow *et al*. [63] introduce joint-aware attention and joint-wise query to predict the entire future sequence without recurrence. Peng *et al*. [48] present a social-aware motion attention mechanism that models both inter- and intra-individual motion relations. DuMMF [73] frames the problem as a dual-level generative task and instantiates it with various generative models. In contrast to previous work, we adopt a MARL perspective and model the strategic social interaction process using cognitive hierarchies.

## 2.4 Cognitive hierarchy theory

Cognitive hierarchy theory (CHT) is a model in behavioral economics and game theory that aims to describe human decision processes in strategic games. Several foundational works introduce the concept of CHT [10], and provide experimental evidence [45, 14, 23, 57, 56] supporting the existence of cognitive hierarchies in human decision-making processes. Taking it a step further, Wright *et al*. [71] develop a Poisson cognitive hierarchy model to predict human behavior in normal-form games. Li *et al*. [35] summarize how the information structure guided by the cognitive hierarchy supports belief generation and policy generation in game-theoretic multi-agent learning. In this work, we propose a cognitive hierarchy framework in conjunction with MARL, and demonstrate its effectiveness in a real-world social motion prediction scenario.

# 3 Dataset

## 3.1 Overview

We present a multi-person 3D motion dataset with a special focus on strategic interactions called `Wusi` (Wusi Basketball Training Dataset). In the following, we first introduce the contents of the dataset, then outline the data collection pipeline, and finally provide statistical analysis and comparison with regard to existing datasets.

Our dataset captures the no-dribble-3-on-2 basketball drills performed by a team of professional basketball athletes. In each drill, three offensive players possess the ball, while another two players

Table 1: Dataset comparison. We compare our dataset with existing multi-person motion datasets employed by previous works on the multi-person motion prediction task. [†] denotes multi-person subset as utilized in previous works [2, 68].

| Dataset | 2D/3D | Frames | Duration (min) | No. of people | Interaction |
|---------|-------|--------|----------------|---------------|-------------|
| PoseTrack [6][†] | 2D | 8K | 5.5 | Multiple | Weak |
| CMU-Mocap [1][†] | 3D | 34K | 4.9 | 2 | Weak |
| 3DPW [65][†] | 3D | 5K | 6.1 | 2 | Weak |
| MuPoTs-3D [41] | 3D | 8K | 4.4 | 2-3 | Weak |
| ExPI [22] | 3D | 30K | 20.0 | 2 | Cooperative |
| **Wusi** | 3D | 60K | 40.3 | 5 | Strategic |

are on defense. The offensive team aims to accomplish as many successful passes as possible within the given time, while the defensive team strives to get steals, deflections, and slow down the offense. Since dribbling is prohibited, the drill requires the offensive players to make better passing decisions and foresee the defense before they pass; conversely, the defensive players need to anticipate the passing directions in order to steal the ball. Figure 1 showcases the diverse and dynamic human interactions present in our dataset. On the left, the offensive player successfully passes the ball to a teammate. In the middle, the defensive player accurately anticipates the passing trajectory and successfully steals the ball. On the right, the offensive player pretends to pass the ball to the right, then swiftly executes a pass to the left.

## 3.2 Data collection

Our motion capture system consists of 11 synchronized and calibrated wide-baseline cameras. Our setup ensures that the players are well-surrounded by the cameras from elevated shooting angles, enabling each body joint to be covered by at least four different camera views. We utilize a markerless multi-person 3D pose estimation algorithm [74] with JDETracker [69]. Subsequently, we apply a post-processing pipeline that includes automatic failure detection and temporal filtering [11]. A final manual check is conducted to guarantee the integrity of the data sequences.

## 3.3 Data analysis

In this section, we examine the statistics of **Wusi** dataset and compare it with existing datasets for multi-person motion prediction. As outlined in Table 1, current datasets exhibit two primary limitations for the multi-person motion prediction task. Firstly, high-quality 3D motion data is limited in terms of scale (duration and the number of people). Consequently, prior works [68, 48, 63, 73] resort to randomly mixing the motion sequences [1, 65, 41] in order to generate more multi-person motion sequences for training. However, such practice inevitably causes unnatural human interactions. Secondly, the interaction strength in existing datasets is restricted, mainly featuring simple actions such as walking together [6, 1, 65, 41]. More recently, the ExPI dataset [22] introduces cooperative 3D motions of two Lindy-hop dancers. In contrast, our dataset offers 2 to 9 times the video duration of existing datasets and includes strategic global interactions among 5 individuals.

Table 2: Comparison of pose diversity based on different thresholds.

| Threshold | 50mm | 100mm |
|-----------|------|-------|
| Human3.6M [28] | 24% | 12% |
| ExPI [22] | 52% | 23% |
| CMU-Mocap [1] | 20% | 9% |
| MuPoTs-3D [41] | 37% | 19% |
| **Wusi** | **53%** | **27%** |

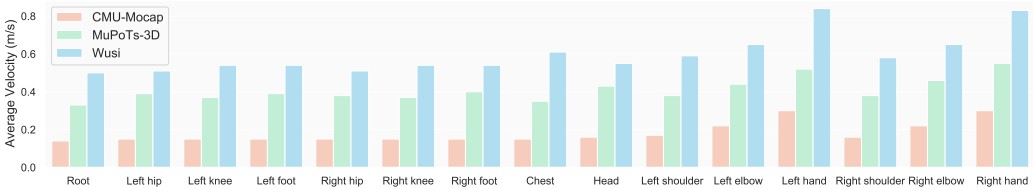

Figure 2: Comparison of motion intensity.

Additionally, we provide a quantitative comparison with existing datasets, emphasizing two critical aspects: *pose diversity* and *motion intensity*. Firstly, we examine the pose diversity within the datasets by calculating the ratio of unique poses to the total number of poses following [28, 22]. Table 2 shows that our dataset surpasses the previous datasets in terms of overall pose diversity. We further analyze the motion intensity by computing the average velocity for all the body joints. As illustrated in Table 2, our dataset exhibits more dynamic movements across all body joints. We suppose that these characteristics make **Wusi** not only challenging for motion prediction, but also potentially useful for other related tasks, *e.g.* motion generation [76].

## 4 Method

### 4.1 Formulation

The multi-person motion prediction problem can be formulated as follows: given the motion histories $\{x_t^p\}_{1 \leq t \leq T, 1 \leq p \leq P}$ of length $T$ from $P$ human subjects, predict their future motion $\{x_t^p\}_{T \leq t \leq T+T', 1 \leq p \leq P}$ of length $T'$, where each $x_t^p$ represents a 3D human pose.

We model the problem using a Markov Decision Process (MDP) [60] by dividing the prediction sequence into $L$ steps with even step length $m = \frac{T'}{L}$ following [66], and further extend the formulation to multiple agents. We use bold symbols to represent the collective variables of all agents following the convention in MARL. At each step $i$, The state $\boldsymbol{s}_i$ is defined as the aggregated motion history $\{x_t^i\}_{1 \leq t \leq T+(i-1) \times m, 1 \leq p \leq P}$ for all the agents. The action for the $p$-th agent $a^p$ is defined as a sequence of velocities $\{v_t^p\}_{T+(i-1) \times m \leq t \leq T+i \times m-1}$, where $v_t^p = x_{t+1}^p - x_t^p$. Therefore, the joint action of all $P$ agents $\boldsymbol{a}_i$ deterministically transition the MDP into the new state $\boldsymbol{s}_{i+1}$ according to:

$$x_{t^*}^p = x_{T+(i-1) \times M}^p + \sum_{t=T+(i-1) \times M}^{t^*} v_t^p, \quad T+(i-1) \times m+ \leq t^* \leq T+i \times m, 1 \leq p \leq P. \quad (1)$$

The goal of this objective is to learn a policy $\pi^p$ for each agent $p$ that maps each possible state $\boldsymbol{s}$ to agent action $a^p$ by $a^p = \pi^p(\boldsymbol{s})$. In the real world, people optimize their $\pi^p$ to maximize an implicit team reward $r$. Since we have no access to $r$ or the environment, we aim to learn $\pi^p$ from expert demonstrations $\mathcal{D}$ via imitation learning.

We parameterize $\pi^p$ with a set of parameters $\theta$. In practice, we follow [68] to employ a global-range Transformer $E_g$ to encode global state feature $\boldsymbol{s}_g = E_g(\boldsymbol{s})$, and a local-range Transformer $E_l$ to encode local state feature $s_l^p = E_l(\boldsymbol{s})$ for each agent $p$. See Figure 3 for an overview. This approach helps to extract multi-level state features and enables all agents to share policy network parameters.

### 4.2 Behavioral Cloning

A straightforward method in imitation learning is Behavioral Cloning (BC) [7, 51], *i.e.*, using expert demonstrations to minimize the difference between the actions produced by the policy and those taken by the experts:

$$\theta^* = \text{argmin}_\theta \sum_{\mathcal{D}} \sum_{p=1}^{P} \ell\left(\pi_\theta^p(\boldsymbol{s}), \tilde{\pi}^p(\boldsymbol{s})\right), \quad (2)$$

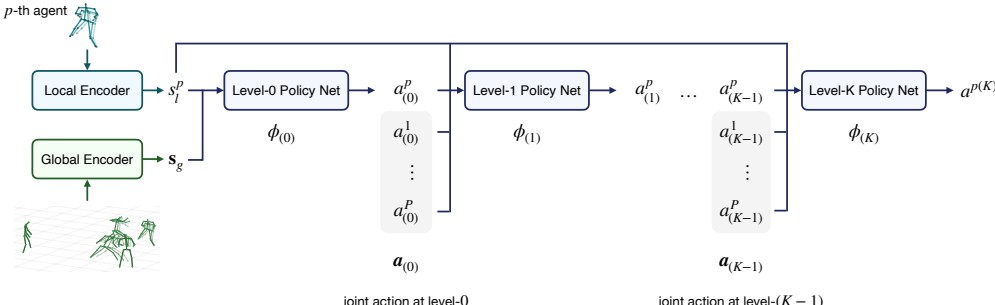

joint action at level-0          joint action at level-$(K-1)$

Figure 3: Framework overview. For the $p$-th agent, two state encoders first extract its local and global state features $s_l^p$ and $s_g$, from which the level-0 policy network produces an initial action $a_{(0)}^p$. The level-$k$ policy network produces action $a_{(k)}^p$ based on $s_l^p$ and the joint actions of the previous level $a_{(k-1)}$ ($k \geq 1$).

where $s$ is sampled from expert demonstrations $\mathcal{D}$, $\pi_\theta^p$ and $\tilde{\pi}^p$ are policies of the model and the expert for agent $p$, $\ell$ is a distance function defined in the action space. While BC enjoys the benefit of being both computation-efficient and sample-efficient [53], it also faces several drawbacks. For instance, policies tend to be overfitted with respect to the state distribution encountered by the experts, leading to suboptimal generalization ability [66, 30]. To cope with this challenge, we further introduce GAIL and cognitive hierachy reasoning.

### 4.3 Generative Adversarial Imitation Learning

Generative Adversarial Imitation Learning (GAIL) [26] is a model-free imitation learning approach. It leverages the adversarial training framework of Generative Adversarial Networks (GAN) [21] and the concept of inferring reward functions from Inverse Reinforcement Learning (IRL) [46]. Direct estimation of reward signals from expert demonstrations can be very difficult; therefore, GAIL transforms the IRL problem into its equivalent dual problem of occupancy measure matching. Specifically, the policy network is regularized to match its state-action pair distribution to that of the experts' policy through adversarial training. We implement GAIL using a global discriminator $D$ parameterized by $\omega$, which differentiates the state-action pairs produced by agents' joint policy $\pi$ and experts' joint policy $\tilde{\pi}$. The optimization objective can be formulated as:

$$\min_\theta \max_\omega \mathbb{E}_{\pi_\theta} \left[ \log D_\omega \left( s, \pi_\theta \left( s \right) \right) \right] + \mathbb{E}_{\tilde{\pi}} \left[ \log \left( 1 - D_\omega \left( \tilde{s}, \tilde{\pi} \left( \tilde{s} \right) \right) \right) \right], \tag{3}$$

where $s$ and $\tilde{s}$ are sampled from expert demonstrations $\mathcal{D}$ independently.

### 4.4 Cognitive hierarchies

Furthermore, we notice that in real-world situations, individuals would predict others' behaviors by forming beliefs about others' policies and then acting accordingly to maximize their own payoffs [20, 9]. This can be formulated as a hierarchical, game-theoretic decision-making process, in which lower-level agents adopt straightforward strategies, while higher-level players anticipate the strategies of lower-level agents and respond accordingly. We explicitly model this recursive reasoning process to learn more interpretable and robust agent policies. We adopt a specific type of cognitive hierarchy model [10] called *level-k thinking* [57, 45, 15] to represent this process. Specifically, agents at each level (except for the lowest one) assume that others are reasoning at the previous level. As shown in Figure 3, a straightforward policy is to take actions based on local and global state features $s_l^p$ and $s_g$, which we denote as level-0 policy $\pi_{(0)}^p$. Level-0 actions are thus defined as:

$$a_{(0)}^p = \pi_{(0)}^p(s) = \phi_{(0)}(s_l^p, s_g), \tag{4}$$

where $\phi_{(0)}$ is the level-0 policy network. Then for a level-$k$ agent ($k \geq 1$), it takes actions based on the joint agent actions $a_{(k-1)}$ at the previous level and its local state features $s_l^p$:

$$a_{(k)}^p = \pi_{(k)}^p(\boldsymbol{s}) = \phi_{(k)}(s_l^p, \boldsymbol{a}_{(k-1)}), \quad 1 \le k \le K \tag{5}$$

where $\phi_{(k)}$ is the level-$k$ policy network, $K$ is the maximum strategic depth.

### 4.5 Training objectives

Finally, we introduce our training objectives which effectively integrate the aforementioned components. For level-$k$ ($1 \le k \le K$) agents, we apply GAIL to regularize the distance between their joint policy and the experts' policy:

$$\mathcal{L}_{\text{GAIL}} = \sum_{k=1}^{K} \mathbb{E}_{\boldsymbol{\pi}_{(k)}} \left[ \log D\left(\boldsymbol{s}, \boldsymbol{\pi}_{(k)}\left(\boldsymbol{s}\right)\right) \right] + \mathbb{E}_{\tilde{\boldsymbol{\pi}}} \left[ \log \left(1 - D\left(\tilde{\boldsymbol{s}}, \tilde{\boldsymbol{\pi}}\left(\tilde{\boldsymbol{s}}\right)\right)\right) \right]. \tag{6}$$

In addition, we employ BC on level-$K$ agent policies:

$$\mathcal{L}_{\text{BC}} = \sum_{\mathcal{D}} \sum_{p=1}^{P} \ell \left( \pi_{(K)}^p \left(\boldsymbol{s}\right), \tilde{\pi}^p \left(\boldsymbol{s}\right) \right). \tag{7}$$

We train the discriminator $D$ and the policy networks $\phi$ alternatively, where $D$ aims to maximize $\mathcal{L}_{\text{GAIL}}$, and $\phi$ is optimized to minimize a linear combination of $\mathcal{L}_{\text{BC}}$ and $\mathcal{L}_{\text{GAIL}}$ using a policy gradient algorithm following [30]:

$$\mathcal{L}_\phi = \mathcal{L}_{\text{BC}} + \lambda \mathcal{L}_{\text{GAIL}}, \tag{8}$$

where $\lambda$ is a constant for balancing the loss terms.

## 5 Experiments

### 5.1 Setup

**Implementation details.** We implement the presented framework to train and test on the proposed `Wusi` dataset. We employ Transformer encoder [62] for both the local and global state encoders, as well as Transformer decoders for the policy networks. Each Transformer consists of 3 layers with 8 attention heads. We share parameters for policy networks $\phi_{(1)} \ldots \phi_{(K)}$. We set the strategic reasoning depth $K = 3$ unless otherwise stated. For all the methods, we provide 1s motion history and predict future 1s motion. For additional experimental details, please refer to the appendix.

**Evaluation metrics.** We compute the Mean Per Joint Position Error (MPJPE) between the predicted future motion and ground-truth. In order to disentangle the results of trajectory prediction and pose prediction, we further calculate the mean root position error and mean local pose error (MPJPE after root alignment). We report all the errors in millimeters.

### 5.2 Evaluation and comparison

In this section, we evaluate the proposed dataset and method, comparing with the prior works. We conduct experiments with the following baseline methods including a naive baseline and multiple state-of-the-art approaches: 1) *Frozen* is a naive baseline that simply replicates the last frame of the input motion. 2) *Social-STGCNN* [42] is a multi-agent trajectory prediction method using spatio-temporal graph convolutional networks. For trajectory prediction methods, we train and test using the root (global) trajectories and replicate the last frame for the root-relative (local) pose. 3) *SocialVAE* [72] is a multi-agent trajectory prediction method using timewise variational autoencoder. 4) *History Repeats Itself (HRI)* [40] is a single-person motion prediction method that allows absolute coordinate inputs. 5) *Multi-Range Transformers (MRT)* [68] predicts multi-person motion utilizing a local-range encoder and a global-range encoder. 6) *Social Motion Transformer (SoMoFormer)* [63] employs the joints of all people as queries and predicts their future motion in parallel.

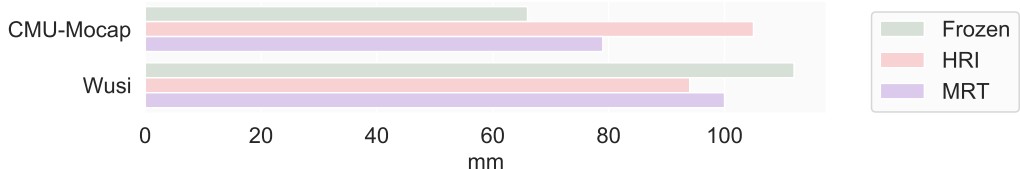

Figure 4: Comparison of baseline methods on CMU-Mocap [1] and our dataset. We compute the 1s mean local pose prediction errors. The baseline methods are trained and tested on two datasets separately.

Table 3: Performance comparison with the baseline methods. We compute the mean prediction errors of global pose, local pose, and root, respectively.

| milliseconds | Global | | | | Local | | | | Root | | | |
|---|---|---|---|---|---|---|---|---|---|---|---|---|
| | 400 | 600 | 800 | 1000 | 400 | 600 | 800 | 1000 | 400 | 600 | 800 | 1000 |
| Frozen | 119.2 | 165.7 | 207.3 | 244.5 | 62.8 | 82.8 | 98.9 | 112.1 | 99.8 | 139.3 | 175.0 | 207.6 |
| Social-STGCNN [42] | 188.2 | 246.4 | 298.8 | 346.5 | 62.8 | 82.8 | 98.9 | 112.1 | 173.1 | 225.6 | 273.7 | 317.9 |
| SocialVAE [72] | 84.0 | 127.2 | 171.7 | 215.1 | 62.8 | 82.8 | 98.9 | 112.1 | 52.6 | 89.7 | 130.3 | 170.8 |
| HRI [40] | **50.0** | 106.8 | 145.5 | 189.2 | **39.5** | 62.9 | 78.4 | 94.1 | **40.1** | 88.0 | 119.6 | 157.7 |
| MRT [68] | 66.9 | 103.2 | 140.2 | 176.4 | 49.4 | 68.8 | 85.4 | 99.5 | 50.6 | 79.0 | 109.3 | 140.1 |
| SoMoFormer [63] | 53.2 | 88.8 | 124.9 | 160.0 | 42.3 | 62.6 | 79.7 | 93.9 | 42.5 | 70.7 | 100.8 | 131.2 |
| Ours | 54.6 | **86.2** | **119.3** | **152.5** | 43.7 | **60.8** | **74.6** | **86.6** | 41.7 | **66.9** | **94.8** | **124.0** |

### 5.2.1 Quantitative results

We first benchmark three baseline methods to assess the motion prediction difficulty of our dataset in comparison to the commonly-used existing dataset [1]. Figure 4 reveals that for the CMU-Mocap dataset [1], merely repeating the last-frame pose (*Frozen*) leads to significantly lower errors than the model predictions. This finding indicates that the motions in this dataset are relatively "static", making it less challenging and informative for the social motion prediction task. In contrast, the dynamic nature of our dataset gives rise to considerably higher prediction errors of the naive *Frozen* baseline. The results demonstrate that the proposed dataset serves as a more convincing benchmark for social motion prediction and introduces more challenges to the field.

Furthermore, we conduct a quantitative evaluation of the proposed framework's performance and compare it with state-of-the-art approaches. As illustrated in Table 3, our method achieves competitive results with SOTA approaches. It is worth noting that the single-person method HRI [40] demonstrates respectable short-term motion prediction accuracy; however, its performance over longer horizons is constrained by the absence of global context, particularly in root trajectory prediction. In contrast, our approach significantly surpasses SOTA methods in long-term social motion prediction, highlighting its effectiveness in modeling long-range human interactions. In the subsequent section, we will demonstrate the critical role that social context plays in predicting long-term human motion via examples.

### 5.2.2 Qualitative results

In order to better understand the performance improvement of our method, we visualize its motion prediction results and compare with the baseline methods qualitatively. Figure 5 illustrates an representative result in which the input motion shows that a defensive player (highlighted) is switching the target of defense. Our approach correctly predicts its future direction and orientations as the offensive player at the other side will likely be covered by the teammate. In contrast, other approaches fail to reason this, predicting that the player would continue to move along the input trajectory or simply stop at the middle.

In Figure 6, three players to be noticed are highlighted in rose, purple and blue (we also call them this way below). Additionally, we visualize the actions produced by the intermediate policy networks to

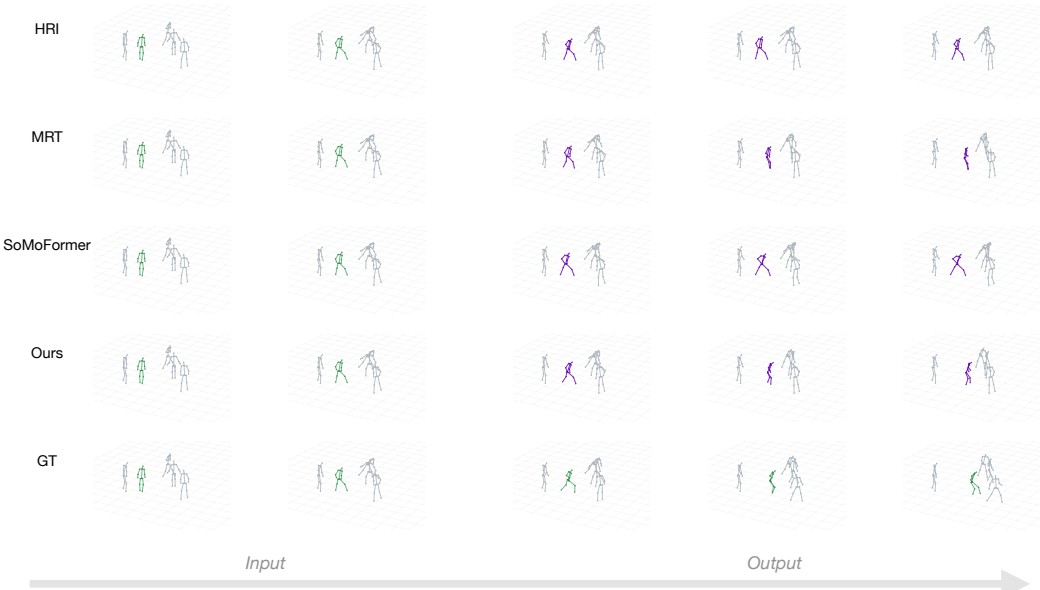

Figure 5: Qualitative comparison with existing methods. Left two columns are past motion and right three columns are future motion. We highlight the prominent player with bright colors. Real motion are shown in green, and model predictions are shown in purple.

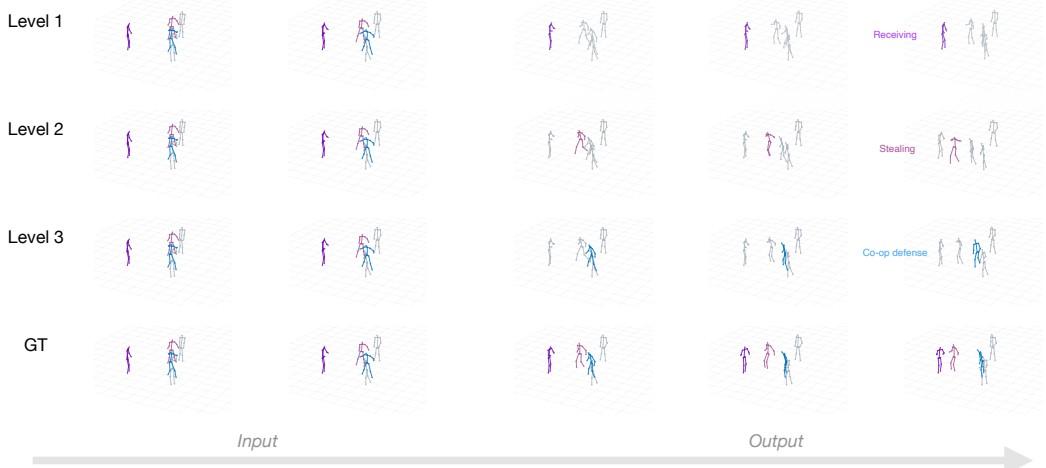

Figure 6: Actions produced by policy networks at different levels. Left two columns are past motions and right three columns are future motions.

interpret the decision-making process. Shown on the top row, player *purple* at level 1 policy network simply follows the past trend to receive the ball. As displayed in the second row, defender *rose* at level 2 gains knowledge from the first level that player *purple* is receiving the ball, and implements a steal attempt. Nevertheless, such aggressive actions could create defensive gaps. Defender *blue* at level 3 senses such gaps caused by defender *rose* from level 2, therefore executes cooperative defense upwards to fill the gap.

## 5.3 Ablation study

Finally, we conduct ablation studies to investigate the influence of each component. In Table 4 (a)-(e), we change the policy network depth $K$. $K = 0$ corresponds to producing the final action directly without any recursive reasoning. We find that introducing the recursive reasoning process ($K > 0$) clearly lowers the prediction errors, proving the effectiveness of our framework design based on cognitive hierarchy. From (f) to (h), we keep $K = 3$ and study the influence of training regularizations. (f) removes the GAIL regularization for intermediate levels $1 \dots K - 1$, and (g)

Table 4: Ablation study of cognition levels and architecture designs.

| milliseconds | Global | | | | Local | | | | Root | | | |
|---|---|---|---|---|---|---|---|---|---|---|---|---|
| | 400 | 600 | 800 | 1000 | 400 | 600 | 800 | 1000 | 400 | 600 | 800 | 1000 |
| (a) $K = 0$ | 60.1 | 94.3 | 129.2 | 163.7 | 47.4 | 66.2 | 82.0 | 95.3 | 45.7 | 72.7 | 101.8 | 131.8 |
| (b) $K = 1$ | 55.2 | 88.0 | 121.9 | 155.5 | 44.3 | 62.1 | 77.0 | 89.7 | 42.3 | 68.0 | 96.3 | 125.4 |
| (c) $K = 2$ | 54.8 | 87.2 | 120.7 | 154.2 | 43.5 | 60.6 | 75.0 | 87.3 | **41.7** | 67.6 | 95.9 | 125.3 |
| (d) $K = 4$ | **54.5** | 86.3 | 119.7 | 153.5 | 43.6 | 60.5 | 74.7 | 86.6 | 42.0 | 67.3 | 95.5 | 125.2 |
| (e) $K = 3$, full | 54.6 | **86.2** | **119.3** | **152.5** | 43.7 | 60.8 | 74.6 | 86.6 | 41.7 | **66.9** | **94.8** | **124.0** |
| (f) *w/o* mid GAIL | 54.9 | 87.5 | 121.4 | 155.4 | 43.7 | **60.4** | **74.5** | **86.4** | 41.8 | 67.8 | 96.6 | 126.9 |
| (g) *w/o* GAIL | 54.6 | 87.1 | 121.1 | 155.1 | **43.4** | 60.4 | 74.7 | 86.8 | 41.9 | 68.1 | 97.0 | 127.0 |
| (h) *w/o* weight sharing | 57.1 | 91.0 | 125.8 | 160.2 | 44.4 | 62.2 | 77.1 | 89.7 | 43.6 | 70.3 | 99.5 | 129.5 |

Table 5: Quantitative comparison of actions produced by policy networks at different levels.

| milliseconds | Global | | | | Local | | | | Root | | | |
|---|---|---|---|---|---|---|---|---|---|---|---|---|
| | 400 | 600 | 800 | 1000 | 400 | 600 | 800 | 1000 | 400 | 600 | 800 | 1000 |
| Level-1 | 76.8 | 122.4 | 170.3 | 217.7 | 50.1 | 69.0 | 85.2 | 99.1 | 61.4 | 100.4 | 143.2 | 186.9 |
| Level-2 | 61.0 | 99.9 | 144.6 | 192.5 | 47.1 | 66.0 | 82.5 | 96.6 | 46.5 | 78.0 | 116.2 | 158.8 |
| Level-3 | **54.6** | **86.2** | **119.3** | **152.5** | **43.7** | **60.8** | **74.6** | **86.6** | **41.7** | **66.9** | **94.8** | **124.0** |

completely removes GAIL. Training without GAIL increases the long-term trajectory prediction errors, while slightly lowers the local pose prediction error. We also observe that models without GAIL tend to produce motions that appear less natural (e.g. limb twisting), leading to inferior visual effects. In (h), we use different policy network instances for levels $1 \ldots K$, and it turns out that sharing the policy networks leads to better generalization.

In addition to the qualitative visualization of actions generated by policy networks at different reasoning levels, we also provide a quantitative comparison of their distance from ground-truth. Table 5 demonstrates that policy networks at different levels learn different actions, and policy networks at higher reasoning levels produce actions with less prediction errors. We hypothesize that, through the proposed recursive reasoning process, policy networks at different levels display a range of cognitive types, with behavior spanning from relatively naive to substantially rational. Higher-level policies more closely approximate an equilibrium strategy, which is usually in alignment with expert demonstrations. We provide several example motion sequences in the video demo to further illustrate this observation.

## 6   Conclusion

In this work, we propose the first large-scale multi-person 3D motion dataset featuring strategic human social interactions. This dataset surpasses the existing ones in scale, diversity, dynamics, and interaction, thus posing new challenges to the social motion prediction problem. We reformulate the problem using a MARL perspective. Building on this, we propose a framework that effectively combines behavioral cloning (BC), generative adversarial imitation learning (GAIL), and cognitive hierarchy. Our approach demonstrates strong generalization capabilities and improved interpretability for modeling strategic human social interactions.

**Limitations:** Our approach is based on the cognitive hierarchy assumption and may not hold true for all type of social interactions. This work do not explicitly model social roles, which can be explored as future work.

**Acknowledgment**   This work was partially supported by National Key R&D Program of China (2022ZD0114900). We thank Xuesong Fan and Rujie Wu for their generous support during the data collection process.

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

# A  Appendix

## A.1  Dataset

### A.1.1  Data Collection

We record basketball drill videos from 11 camera views at a frame rate of 25 FPS. Multi-person 3D poses are estimated using Faster-VoxelPose [74]. To obtain 2D heatmaps, we first resize the image to $960 \times 512$ pixels and employ the ResNet [24] backbone. The 3D space size is set to $16m \times 16m \times 3m$, with the granularity of human detection networks at $80 \times 80 \times 20$. To estimate the joint locations of each subject, we set the human bounding box size to $2m \times 2m \times 2.4m$ and use a voxel size of $64 \times 64 \times 64$. We further apply JDETracker [69] with a tracking frame rate of 25FPS and a threshold of $0.2m$. Next, we conduct an automatic failure detection process. We first discard motion clips with incorrect person counts and then filter outliers with root position changes exceeding a $0.4m$ threshold compared to neighboring frames. Subsequently, we adopt the one-euro filter [11] for temporal smoothing. We set different parameters for different body joints because they exhibit different moving patterns. For upper limb joints such as wrists and elbows, we set $f_{\text{cutoff}} = 0.0287$ and $\beta = 0.5$; for lower body joints including ankles and knees, we set $f_{\text{cutoff}} = 0.0274$, $\beta = 1$; and for the rest of the body, we set $f_{\text{cutoff}} = 1$, $\beta = 1$. Finally, we perform a manual check to ensure the correctness of the data sequences.

### A.1.2  Data Analysis

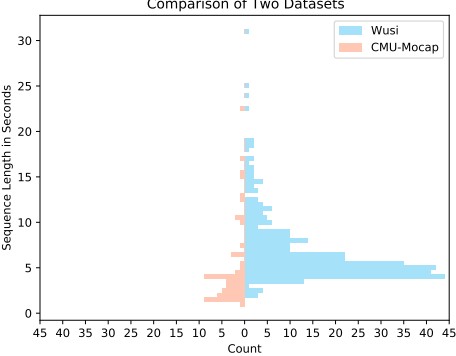

Figure 7: Clip length distribution comparison between our dataset and CMU-Mocap [1].

We visualize the clip lengths of our dataset and compare them with CMU-Mocap [1]. Figure 7 shows that our dataset features a longer overall duration and a larger number of long sequences.

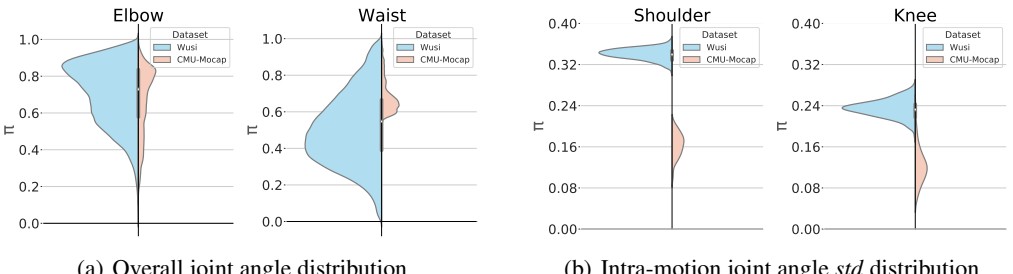

(a) Overall joint angle distribution          (b) Intra-motion joint angle *std* distribution

Figure 8: Distribution comparison between our dataset and CMU-Mocap [1]. (a): We plot the joint angle distribution over the entire dataset. (b): We calculate the joint angle standard deviation for each motion sequence, then plot the *std* distribution.

Furthermore, we assess pose diversity and motion intensity utilizing joint angle distributions. As Figure 8(a) shows, our dataset encompasses a broader range of plausible body joint angles, indicating a

superior pose diversity. We then investigate motion intensity using intra-motion variation. Figure 8(b) illustrates that motion sequences in our dataset have higher joint angle variances, indicating more dynamic motions.

## A.2 Implementation details

We represent human motion with 3D skeleton sequences following previous works [68, 48, 55]. Specifically, each $x_t^p \in \mathbb{R}^{J \times 3}$ represents a 3D human pose for person $p$ at timestep $t$, containing the Cartesian coordinates of $J$ skeleton joints.

We apply Discrete Cosine Transform (DCT) [4] to the input motions and Inverse Discrete Cosine Transform (IDCT) for the decoded actions following [40, 68]. We employ a local-range Transformer encoder and a global-range Transformer encoder [68] as the state encoders. We utilize Transformer decoders as the policy networks. We concatenate the local state features $s_l^p$ with global state features $s_g$ (for level-0) or joint agent actions at the previous level $a_{(k-1)}$ (for level-$k$, $k \geq 1$). For GAIL training, we implement a discriminator $D$ that takes a pair of state $s$ and joint actions $a$ as input and aims to classify whether the actions are from the policy network or expert demonstrations. The discriminator consists of a global-range Transformer encoder, a temporal average pooling layer, and an MLP with one hidden layer. We implement the proposed framework with PyTorch [47] using a Linux machine with 1 NVIDIA V100 GPU. We set $\lambda = 0.002$, batch size 32, learning rate 0.0001, and train for 60 epochs using Adam [33] optimizer.

## A.3 Ablation Study

Table 6: Ablation study of $\lambda$.

| milliseconds | Global | | | | Local | | | | Root | | | |
|---|---|---|---|---|---|---|---|---|---|---|---|---|
| | 400 | 600 | 800 | 1000 | 400 | 600 | 800 | 1000 | 400 | 600 | 800 | 1000 |
| $\lambda = 0.01$ | 61.3 | 94.1 | 127.5 | 160.2 | 50.3 | 69.3 | 84.9 | 97.4 | 45.6 | 71.1 | 99.2 | 128.0 |
| $\lambda = 0.005$ | 54.8 | 86.9 | 120.5 | 154.6 | 43.8 | 61.1 | 74.9 | 87.4 | 41.8 | 67.4 | 97.6 | 126.2 |
| $\lambda = 0.002$ | 54.6 | 86.2 | **119.3** | **152.5** | 43.7 | 60.8 | 74.6 | 86.6 | 41.7 | 66.9 | **94.8** | **124.0** |
| $\lambda = 0.001$ | 54.3 | **86.1** | 119.5 | 153.5 | 43.1 | 60.0 | **74.2** | **86.1** | **41.4** | **66.8** | 95.2 | 125.3 |
| $\lambda = 0.0002$ | **54.2** | 86.5 | 120.2 | 154.1 | **43.0** | **59.9** | 74.2 | 86.2 | 41.6 | 67.6 | 96.2 | 126.1 |

We provide an additional ablation study on loss weight $\lambda$ in Table 6.

## A.4 Discussions

**Broader Impacts** Our approach does not rely on explicit biometric markers like unique fingerprints, retina scans, or body markers. The skeletal structure representation utilized in our dataset is based on generalized joint positions and movements. Our approach should only be employed with legally obtained human motion data. Meanshile, our approach may contribute to more interpretable decision-making processes and holds potential for applications in sports tactical analysis and education.

**Assets** To collect our dataset, we acquire written consent from the participating athletes and coaches. We plan to release the processed 3D human motion sequences under a license agreement that permits their use for non-commercial scientific research purposes.

**Human Subjects** We inform the participating athletes about the data collection process and its intended uses, ensuring that we obtain their written consent. The instructions are provided by their coach, and the collected drills are part of their regular training routines. All participants receive compensation in accordance with their standard practices. To ensure privacy, we have implemented rigorous measures. Our data release strategy does not involve disclosing raw video data that may contain identifiable information, such as facial features. Instead, we will exclusively release processed 3D human motion sequences derived from the video data, guaranteeing the anonymity of the athletes.

