# OpenReview forum: "Social Motion Prediction with Cognitive Hierarchies"
_NeurIPS.cc/2023/Conference — NeurIPS 2023 poster_

### Official Review · Reviewer_yiis · 2023-06-09

**Soundness:** 3 good
**Presentation:** 2 fair
**Contribution:** 2 fair
**Rating:** 5
**Confidence:** 4

**Summary:**

This paper proposes a novel approach to address the social motion prediction problem by introducing a new large-scale multi-person 3D motion dataset featuring intense and strategic interactions among participants. The authors formulate the problem using a multi-agent reinforcement learning perspective and incorporate behavioral cloning and generative adversarial imitation learning to boost learning efficiency and generalization. They also take into account the cognitive aspects of the human social action planning process and develop a cognitive hierarchy framework to predict strategic human social interactions. The proposed approach outperforms state-of-the-art methods in challenging long-term social motion predictions.


**Strengths:**

The main strengths of this paper include:
+ Introducing and analyzing the limitations of existing datasets for multi-person motion prediction tasks.
+ Proposing the use of GAIL to improve the generalization ability of the learned policies.
+ Provide a demo video.

**Weaknesses:**

**Good motivation and interesting methods, but not sufficient efforts to support the claim.** I will state weaknesses as follows:


+ Distribution comparison in Fig. 2 shows the comparison with CMU-Mocap, which is an old dataset. For claiming the dataset novelty, I suggest authors compare your dataset with more datasets here. Only several sub-figures of several joints are not sufficient to reflect the distribution of the full dataset.
+ For dataset comparison, [1] has 2.2 hours of data and [2] is 6.5 hours. These two datasets should be compared.
+ **Over-claim: I think a dataset of less than an hour cannot claim to be a large-scale dataset.**
+ In lines 29-32, why [56, 1, 25] cannot be used for interactive tasks? CMU-Mocap can be used to perform motion prediction and generation tasks.
+ Authors mentioned NTU-RGBD in Section 2.1 but did not compare with it in Table 1.
+ Writing:
  + orphan row (lines 233, 251, Table 3).
  + Line 236: Table 4 -> Figure 4




[1]: Umpm benchmark: A multi-person dataset with synchronized video and motion capture data for evaluation of articulated human motion and interaction.

[2]: InterGen: Diffusion-based Multi-human Motion Generation under Complex Interactions.

**Questions:**

**According to review guidelines, 'if the contribution is a dataset or model, what steps did you take to make your results reproducible or verifiable?' The authors did not claim it in the main paper or abstract (It is important. Do NOT claim it in the appendix). Will authors only make a subset of data public or a full set?**

+ Any ablation of $\lambda$s?
+ Multi-person motion capture datasets can be used not only for prediction but also for motion generation. Recent research (HumanMAC: Masked Motion Completion for Human Motion Prediction) use generative frameworks for motion prediction, and it will be great to discuss motion generation.
+ w/o GAIL and w/o mid GAIL ablation shows that the design works little for global pose and local pose. I would like to discuss this with the authors.
+ "This dataset surpasses the existing ones in scale, diversity, dynamics, and interaction, thus posing new challenges to the social motion prediction problem. " Little experiments to verify how the (1)scale, (2)diversity, and (3)interaction benefit the task?
+ Description of annotation, SMPL or SMPL-X? How many joints?
+ For motivation, why do authors build a dataset with intense, strategic interactions? Why are intense, strategic interactions essential for the task and the community? Intense, strategic interactions property is a characteristic of your dataset, but a lack of motivation. It will be great to show how these properties benefit the task in experiments.

**Although I have many concerns about the dataset, the author's method of technological innovation is still interesting. I provide a borderline score here. If the author could discuss and address my concerns, I would improve my score.**


**Limitations:**

Limitations are discussed.

---

> ### Author Rebuttal · Authors · 2023-08-10
>
> We sincerely appreciate the reviewer's constructive and positive feedback. We are grateful that you are interested in our 'method of technological innovation' with 'good motivation'. In the following, we seek to address your concerns:
>
> **Q1: Compare with more datasets and joints.**
>
> A1: We primarily compare with CMU-Mocap [1] as it is the main dataset used by previous works in this area. We will add more dataset comparisons with other datasets and joints following your suggestion.
>
> **Q2: UMPM and InterHuman dataset.**
>
> A2: Please refer to General Response Q1 for comparisons. We will cite and compare with the datasets following your suggestion.
>
> **Q3: Overclaim data size.**
>
> A3: Thanks for your valuable feedback on our dataset size claim. Nonetheles, it is important to note that the dataset is larger than any previously used in the field of multi-human motion prediction. Our dataset provides 3D skeleton ground truths with intense and strategic human interactions from professional atheletes. From such a perspective, we have greatly enlarged the available dataset in this research field for academic uses. We understand how the term "large-scale" might be misconstrued given the temporal length of the dataset. Therefore, in our revised manuscript, we will carefully modify the wording to more accurately represent the size and value of our dataset.
>
>
> **Q4：Why cannot be used for interactive tasks.**
>
> A4: While it's true that these datasets can be used for motion prediction and generation tasks, their utility for tasks involving interaction - a key focus of our research - is limited. This is due to the relatively low level of interactiveness, as we mentioned in Line 30-32. For instance, the individuals in these datasets are often recorded while moving casually or interacting randomly with others, rather than participating in strategic or purposeful social interactions.
>
> **Q5：NTU-RGB+D.**
>
> A5: NTU-RGB+D contains some action classes with two interacting humans. We provide a comparison in Table 3 of the rebuttal PDF. We will add the comparisons to Table 1 in the revision.
>
> **Q6:  Dataser release.**
>
> A6: The entire dataset, as well as our code and models, will be publicly released.
>
> **Q7: Ablation of $\lambda$s.**
>
> A7: We provide an ablation study on $\lambda$ in Table 2 of the rebuttal PDF.
>
> **Q8: Discuss motion generation.**
> A8: Thanks for your suggestion, and our dataset can also be potentially used for motion generation. We will cite and discuss HumanMAC and other related works on motion generation in _Introduction_ and _Related Works_ sections of our revised manuscript.
>
> **Q9: GAIL effects.**
>
> A9: Thanks for your insightful comment. In our manuscript, we have briefly addressed this observation in Line 275-276. Training without GAIL tends to increase the errors in long-term trajectory predictions, while it marginally reduces the errors in local pose prediction. Interestingly, we notice that models trained without GAIL tend to generate motions with smaller magnitudes, often resembling a "freezing" effect. This results in visually less appealing and less realistic animations. Moreover, an important aspect to note is the impact of GAIL on the emergence of cognitive hierarchies within the model. In our experiments, the absence of GAIL leads to a lack of discernible cognitive hierarchy.
>
> **Q10: Verify how the (1)scale, (2)diversity, and (3)interaction benefit the task?**
>
> A10: Thank you for bringing up the important point about the benefits of the scale, diversity, and interaction of our dataset for the social motion prediction task. We are pleased to clarify these aspects:
>
> - Scale: Larger datasets are generally considered beneficial for research tasks as they provide a more extensive base for training models, improving their ability to generalize to unseen data. The advantageous size of our model makes it a valuable resource of the task for the research community.
>
> - Diversity: As shown in Figure 4 and discussed in Line 235-242, our dataset exhibits a high degree of diversity. The results demonstrate that the proposed dataset serves as a more convincing benchmark for social motion prediction and introduces more challenges to the field.
>
> - Interaction: As evidenced by Table 2 and Line 245-249, while the HRI [32] model performs exceptionally well on short-term motion prediction, the global context becomes crucial for long-term motion prediction. Our dataset, with its highly interactive nature, offers strong cues for motion prediction with a global context. These interactions are vital to understanding and predicting complex social behaviors.
>
> We appreciate your feedback, and will revise our manuscript to articulate these points more clearly.
>
>
> **Q11: Annotation.**
>
> A11: Please refer to General Response Q2. We will make it more clear in the revision.
>
> **Q12: Motivation for the dataset.**
>
> A12: Thank you for your insightful question.
>
> - Our primary motivation in focusing on intense, strategic interactions stems from a desire to more accurately capture the complexity inherent in real-world human interactions. These interactions often involve strategic decision-making processes and can be social, competitive, or cooperative in nature. Most existing datasets do not adequately capture these complexities, which is a gap that our dataset aims to fill.
> - By incorporating such interactions, we enable the model to learn beyond simple human motion patterns. Our dataset encourages a deeper understanding of human actions, including aspects such as intentions and the decision-making process. Ultimately, our goal is to develop models that not only understand human behaviors but can also interact meaningfully with humans.
> - As for how these properties benefit our task, we refer you to discussions in A10 (also Table 4, Figure 2, Line 235-249).
>
> We will revise our presentation to better highlight how these properties of our dataset contribute to the task and benefit the broader research community.

---

> > ### Comment · Reviewer_yiis · 2023-08-13
> >
> > Thanks for the authors' feedback.
> >
> > + For A1,
> > > We will add more dataset comparisons with other datasets and joints following your suggestion.
> >
> >     Please detail what you will revise, which will make your promise more convincing.
> >
> > + For A3, similar to my response to A1, please compare all multi-person datasets in detail, which will make claims more fair.
> >
> > + For A4. Authors might revise the statements in the manuscript.
> >
> > + For A5-A8, will you release your codes about all experiments in the paper, rebuttal, and appendix? Note that your promise will be open if accepted.
> >
> > + For `models trained without GAIL tend to generate motions with smaller magnitudes, often resembling a "freezing" effect.`, any metrics to reflect it? I think a freezing score (velocity lower than xx m/s for yy seconds) is needed to verify it. It is better to provide an anonymous demo link to verify it.
> >
> > + For scale, diversity, and interaction properties, if it is not as you claim, please revise it. If the task really benefits from the scale and diversity, I suggest providing more experiments in review, but the authors seem to ignore it. If it is confusing, I take an example here. For the existing relatively small dataset A, you train it and zero-shot test it on your dataset. For your dataset, you train it and zero-shot test it A dataset. This verifies the dataset scaling contribution of your work.
> >
> > + For annotation, why do you use 15 joints and visualize in the mesh? Do you use Smplify? Playing basketball is highly related to hand motion, how can only 15 joints reflect it? I think it might not be physically reasonable. Besides, does your dataset include ball annotation?
> >
> > + For all writing concerns, I suggest authors list all of them for checking.
> >
> >
> > Overall, I will maintain my score now. My existing concerns will affect my final rating.

---

> > > ### Author Response · Authors · 2023-08-14
> > >
> > > We appreciate your recent feedback and suggestions. Below, we present our responses to your insights.:
> > > 1. (A1) We plan to compare with two additional datasets, ExPI [19] and Mupots3D [33], and plot all the major body joints.
> > > 2. (A3) Please refer to [General Response](https://openreview.net/forum?id=lRu0dN7BY6&noteId=u8w2zSFa4e) (Q1) and Rebuttal Table 3. We will add them to Section 2.1 and Table 1.
> > > 3. (A4) We will revise line 29-32 accordingly.
> > > 4. (A5-A8) Yes, all experiments in the paper, rebuttal, and appendix will be included in the open source code.
> > > 5. (GAIL) Thanks for your suggestion. Time permitting, we plan to include additional metric comparisons and a video in the comments; otherwise, these enhancements will be incorporated in the revised manuscript.
> > > 6. (Scale, diversity, and interaction) We appreciate your insightful suggestion. To further emphasize our point, we will incorporate additional experiments beyond those displayed in Table 4 and Figure 2, and refine our presentation to more effectively highlight the unique properties of our dataset. However, it's crucial to clarify that our intention is not to assert that our dataset contributes by *benefiting the existing benchmarks due to its larger scale*. Instead, we posit that it *introduces significant new challenges in a novel area*, specifically the prediction of strategic human interactions as outlined in A12. Our dataset differs markedly from existing ones in terms of motion content (i.e., basketball tactics versus walking, talking, or dancing). Consequently, we don't anticipate existing methods to readily generalize across these markedly different datasets.
> > > 7. (Annotation) Please also refer to [General Response](https://openreview.net/forum?id=lRu0dN7BY6&noteId=u8w2zSFa4e) (Q2) and [Response to Reviewer X2WW](https://openreview.net/forum?id=lRu0dN7BY6&noteId=ED3o4BTKWg) (Q5).
> > >    * We use 15 joints for fair comparison with existing methods [59, 19, 54, 57, 40, 32] following their practice.
> > >    * We use an [existing tool](https://github.com/wangsen1312/joints2smpl) to fit SMPL model using 3D joints. SMPLify fits SMPL using 2D joints.
> > >    * We agree that hand motion is important for studying sport motions. Nonetheless, it is a challenging and unresolved task to accurately recover hand motion in a markerless setting given the small size of the hands, their fast movement, and ball-hand occlusions, which should be further investigated in future work. Meanwhile, our decision to utilize 15 joints (same as previous works) in our model is driven by their ability to adequately capture the broader aspects of human movement, and effectively reflect the interactive properties inherent in our dataset.
> > > 8. (Writing) We have compiled a summary of the writing-related issues that need to be addressed during the revision process, provided below for your reference:
> > >    * Introduction: revise line 29-32, add CHT overview
> > >    * Related Work: add more multi-person datasets, add trajectory prediction datasets and methods, discuss motion generation (e.g. HumanMAC)
> > >    * Dataset: revise claims, add more visualization comparisons, add annotation details and motivation, add datasets to Table 1, revise Table 1 dataset name and sequences
> > >    * Experiments: orphan row (lines 233, 251, Table 3), Line 236: Table 4 -> Figure 4, add GAIL ablation study comparisons

---

> > > > ### Comment · Reviewer_yiis · 2023-08-15
> > > > **Re: Official Comment by Authors**
> > > >
> > > > + For (1), the "we plan to" claim is not convincing. Please provide results and promise to add these results to the paper if accepted.
> > > >
> > > > + For (2), where is the InternHuman dataset in Table 3?
> > > >
> > > > + For (5), not convincing. Please present the results.
> > > >
> > > > + For (6), "we don't anticipate existing methods" is not convincing. No experiments to verify it. I have suggested how to verify it, but authors acknowledge it and did not do it. For such a claim, you need to verify it. Otherwise, it is overclaiming.
> > > >
> > > > + Previous work may be limited by data collection conditions and cannot collect high-quality data. However, it will be meaningless if the work on a dataset has been following previous settings with limited new challenges.
> > > >
> > > > + Using joint2smpl introduce errors. Why not capture the SMPL(-X) data?
> > > >
> > > > + Data with only 15 joints is difficult to use.
> > > >
> > > > + Please read my "I read other reviews and comment here" reply.
> > > >
> > > > Finally, my main concern now is that the authors only claim to provide experimental comparisons, but I can't see these experiments at the moment. This makes it difficult to convince me and also to verify the author's claim. Besides, I am also concerned about the data format.

---

> > > > > ### Author Response · Authors · 2023-08-21
> > > > > **[1/2] More results and discussions**
> > > > >
> > > > > To help address your remaining concerns, we provide more experimental comparisons and dataset discussions as follows:
> > > > >
> > > > > ## Experimental Comparisons
> > > > >
> > > > > **Q(1): I suggest authors compare your dataset with more datasets here. Only several sub-figures of several joints are not sufficient to reflect the distribution of the full dataset.**
> > > > >
> > > > > A(1): Following your suggestions, we have conducted a more comprehensive analysis of dataset comparisons considering **more datasets**, **all the joints**, and using **quantitative metrics**.
> > > > >
> > > > > 1. **Pose diversity.** We follow [A, 19] to analyze the diversity of the datasets by calculating the number of distinct poses. The diversity of the dataset is then defined as the percentage of distinct poses among all the poses. As shown in the table below, our dataset surpasses the previous datasets in terms of overall pose diversity.
> > > > >
> > > > > | Threshold     | 50mm    | 100mm   |
> > > > > | ------------- | ------- | ------- |
> > > > > | Human3.6M [A] | 24%     | 12%     |
> > > > > | ExPI [19]     | 52%     | 23%     |
> > > > > | CMU-Mocap [1] | 20%     | 9%      |
> > > > > | Mupots3D [33] | 37%     | 19%     |
> > > > > | Ours     | **53%** | **27%** |
> > > > >
> > > > > 2. **Motion intensity.** We calculate the average velocity for all body joints (m/s). As shown in the table below, our dataset exhibits more intense movements across all body joints.
> > > > >
> > > > > | Joint    | Root    | Left hip | Left knee | Left foot | Right hip | Right knee | Right foot | Chest    | Head     | Left shoulder | Left elbow | Left hand | Right shoulder | Right elbow | Right hand |
> > > > > | -------- | ------- | -------- | --------- | --------- | --------- | ---------- | ---------- | -------- | -------- | ------------- | ---------- | --------- | -------------- | ----------- | ---------- |
> > > > > | CMU-Mocap [1]    | 0.14    | 0.15     | 0.15      | 0.15      | 0.15      | 0.15       | 0.15       | 0.15     | 0.16     | 0.17          | 0.22       | 0.30       | 0.16           | 0.22        | 0.30        |
> > > > > | Mupots3D [33]   | 0.33    | 0.39     | 0.37      | 0.39      | 0.38      | 0.37       | 0.40        | 0.35     | 0.43     | 0.38          | 0.44       | 0.52      | 0.38           | 0.46        | 0.55       |
> > > > > | **Ours** | **0.50** | **0.51** | **0.54**  | **0.54**  | **0.51**  | **0.54**   | **0.54**   | **0.61** | **0.55** | **0.59**      | **0.65**   | **0.84**  | **0.58**       | **0.65**    | **0.83**   |
> > > > >
> > > > > We will incorporate these quantitative comparisons into Section 3.3 of the revised manuscript.
> > > > >
> > > > > **Q(5): GAIL effects.**
> > > > >
> > > > > A(5): We further analyze the impact of GAIL following your suggestions. Specifically, we found that models trained without GAIL tend to predict motions that appear less natural (e.g. limb twisting). Please refer to the video results (privately submitted to AC following the instructions) for comparison. We will revise the manuscript accordingly to reflect these findings.
> > > > >
> > > > > **Q(6): For the existing relatively small dataset A, you train it and zero-shot test it on your dataset. For your dataset, you train it and zero-shot test it A dataset.**
> > > > >
> > > > > A(6): We perform cross-dataset test experiments following your suggestions. We train and test MRT [59] on CMU-Mocap [1] and our dataset. The table below shows the prediction errors for 1 second.
> > > > >
> > > > > |           | Global         | Local          | Root           |
> > > > > | --------- | -------------- | -------------- | -------------- |
> > > > > | Ours->CMU | 311.2          | 165.3          | 255.2          |
> > > > > | CMU->Ours | 415.6 (+33.5%) | 218.5 (+32.2%) | 353.2 (+38.4%) |

---

> > > > > > ### Author Response · Authors · 2023-08-21
> > > > > > **[2/2] More results and discussions**
> > > > > >
> > > > > > ## Dataset Discussions
> > > > > > **Q(2): InterHuman Dataset.**
> > > > > >
> > > > > > A(2): We initially omitted the InterHuman Dataset from the table for two reasons: 1) this is a concurrent work that has not been published yet; 2) it was neither designed for, nor utilized in, the multi-person motion prediction task. However, for your convenience, we have now included it in the table below.
> > > > > >
> > > > > > | Dataset                     | 2D/3D | Sequences | Frames | Duration (min) | No. of people | Interaction |
> > > > > > | --------------------------- | ----- | --------- | ------ | -------------- | ------------- | ----------- |
> > > > > > | KTH Multiview Football [R8] | 2D&3D | 5         | 6.9k   | 3.9            | 2-3           | None        |
> > > > > > | Haggling [R13]              | 3D    | 34        | -      | 55.6           | 3             | Medium      |
> > > > > > | UMPM (Interactive) [R1]     | 3D    | 8         | 88k    | 29.6           | 2-4           | Medium      |
> > > > > > | NTU-RGB+D (SoMoF) [47]      | 3D    | 739       | -      | 28.2           | 2             | Medium      |
> > > > > > | InterHuman [R14]      | 3D    | 6022       | 1.4m      | 393.6           | 2             | Medium      |
> > > > > > | Ours                        | 3D    | 339       | 60k    | 40.3           | 5             | Strategic   |
> > > > > >
> > > > > > **Q(7): Dataset format.**
> > > > > >
> > > > > > A(7): We first revisit our decision to use 3D keypoints as the data format, as discussed in [General Response](https://openreview.net/forum?id=lRu0dN7BY6&noteId=zGrKNoOyqi) (Q2):
> > > > > >
> > > > > > - **From a methodological standpoint**, most existing works in this domain [59, 19, 54, 57, 40, 32] employ 3D body keypoints in their models.
> > > > > > - **From a data collection standpoint**, marker-based motion capture is more suitable for our specific use case, which involves intense sports training. The state-of-the-art, markerless multi-view motion capture methods [64, R15, R16] primarily aim at estimating 3D body keypoints.
> > > > > >
> > > > > > Then, we respond to the remaining questions:
> > > > > >
> > > > > > 1. **Q: "It will be meaningless if the work on a dataset has been following previous settings with limited new challenges."**
> > > > > >
> > > > > >    A: The new challenge of our dataset lies in the intense, strategic human interactions as elaborated in the manuscript and in A12 of the [rebuttal](https://openreview.net/forum?id=lRu0dN7BY6&noteId=tXSeKDCLUx).
> > > > > >
> > > > > > 2. **Q: "Why not capture the SMPL(-X) data?"**
> > > > > >
> > > > > >    A: To accurately capture body shape and rotations for SMPL(-X), existing works [B] rely heavily on marker-based motion capture techniques, including reflective markers and Inertial Measurement Units (IMUs). Despite their precision, the marker-based solutions are not suitable for our use case (basketball training) because 1) The wearable mocap markers could restrict the athletes' movements and negatively impact their performance; 2) During rapid movements and physical confrontations, the stability of the markers' placement cannot be guaranteed. Therefore, markerless motion capture methods are generally deemed more appropriate for this kind of dataset [19].
> > > > > >
> > > > > > 3. **Q: "Data with only 15 joints is difficult to use."**
> > > > > >
> > > > > >    A: Wang _et al._ [59] aligns the pose representation of four datasets [1, 56, 33, 25] to a unified 15-joint representation, and is followed by other recent works [40, 54, 63]. **Therefore, we adopt the same keypoint definition so that future researchers in this field can seamlessly switch between the datasets and easily compare the datasets and methods.** Additionally, there are a number of widely-used datasets with a similar number of joints: PennAction [C] (13 joints), Campus [D] (14 joints), Shelf [D] (14 joints), Panoptic [25] (15 joints), PoseTrack [5] (15 joints), MPII [E] (16 joints), and Human3.6M [A] (17 joints).
> > > > > >
> > > > > >
> > > > > > **References**
> > > > > >
> > > > > > [A] Ionescu, Catalin, et al. "Human3. 6m: Large scale datasets and predictive methods for 3d human sensing in natural environments." T-PAMI 2013.
> > > > > >
> > > > > > [B] Mahmood, Naureen, et al. "AMASS: Archive of motion capture as surface shapes." ICCV 2019.
> > > > > >
> > > > > > [C] Zhang, Weiyu, Menglong Zhu, and Konstantinos G. Derpanis. "From actemes to action: A strongly-supervised representation for detailed action understanding." ICCV 2013.
> > > > > >
> > > > > > [D] Belagiannis, Vasileios, et al. "3D pictorial structures for multiple human pose estimation." CVPR 2014.
> > > > > >
> > > > > > [E] Andriluka, Mykhaylo, et al. "2d human pose estimation: New benchmark and state of the art analysis." CVPR 2014.

---

> > > > > > > ### Comment · Reviewer_yiis · 2023-08-21
> > > > > > > **Re: More results and discussions**
> > > > > > >
> > > > > > > Dear authors,
> > > > > > >
> > > > > > > I have only seen the authors' responses to my question so far. I do not see responses from other reviewers now. Is there a bug in the system again? Or, did the authors not hear back from other reviewers?
> > > > > > >
> > > > > > > Since the author submitted multiple responses, I needed to double-check. Owing to time constraints, I may not be able to reply to the authors before the end of the author-reviewer discussion. Hope the author can understand. I will also fully participate in the reviewer-AC discussions.
> > > > > > >
> > > > > > > Best,
> > > > > > >
> > > > > > > Reviewer yiis

---

> > > > > > > > ### Author Response · Authors · 2023-08-21
> > > > > > > > **Re: Re: More results and discussions**
> > > > > > > >
> > > > > > > > Dear Reviewer yiis,
> > > > > > > >
> > > > > > > > Thank you for your prompt response.
> > > > > > > >
> > > > > > > > We’ve not yet received responses from other reviewers. We are in communication with the Area Chairs to check if there might be a visibility issue that needs to be addressed.
> > > > > > > >
> > > > > > > > Should you have any immediate queries or require further clarification on any aspects of our paper, we are more than willing to discuss and provide additional information. Your understanding and insights are greatly valued.
> > > > > > > >
> > > > > > > > We deeply appreciate your time and effort in reviewing our work.
> > > > > > > >
> > > > > > > > Best Regards,
> > > > > > > >
> > > > > > > > Authors

---

> > > > > > > ### Comment · Reviewer_yiis · 2023-08-22
> > > > > > >
> > > > > > > Thanks to authors' efforts.
> > > > > > >
> > > > > > > A1, A2, and A6 resolved my concerns. I list my remaining concerns.
> > > > > > >
> > > > > > > - For the nonfreezing claim, I suggested authors evaluate the freezing score:
> > > > > > > > For `models trained without GAIL tend to generate motions with smaller magnitudes, often resembling a "freezing" effect.`, any metrics to reflect it? I think a freezing score (velocity lower than xx m/s for yy seconds) is needed to verify it. It is better to provide an anonymous demo link to verify it.
> > > > > > >     Authors use velocity to replace it. I think velocity reflects responsiveness more, and the freezing score reflects over-smoothing (freezing) problems. They are not the same. Therefore, my concern still exists.
> > > > > > >
> > > > > > > - Authors claim that `We use 15 joints for fair comparison with existing methods [59, 19, 54, 57, 40, 32] following their practice.`. Fair comparisons in scientific problems are primarily comparisons of methods. The captured data format of datasets should be closely related to the tasks claimed by the authors. Since the data collected in this work is closely related to playing basketball, I still think it is difficult to reflect the 15 key-points. In addition, the 15 key-points will also have certain errors when restoring the mesh. Moreover, it's hard to keep the physics of motions and balls plausible. These concerns can lead to difficulties in using the dataset.
> > > > > > >
> > > > > > > - It is not yet clear to me the authors' motivation for choosing action prediction as an example task. Besides, I suggest add the experiments on motion generation.
> > > > > > >
> > > > > > > These are my current concerns, and I will discuss related concerns with other reviewers and AC in the following discussion phase.

---

> > > > > > > > ### Author Response · Authors · 2023-08-22
> > > > > > > > **Addressing remaining concerns**
> > > > > > > >
> > > > > > > > Thank you for acknowledging the resolution of some of your concerns and for detailing the remaining ones. We provide additional clarification below to address these points.
> > > > > > > >
> > > > > > > > 1. **Authors use velocity to replace it.**
> > > > > > > >
> > > > > > > > The computation of per-joint velocity is intended to depict the intensity of motion within the dataset rather than to make comparisons with the GAIL ablation version. As mentioned in the response to A(5), models trained without GAIL tend to predict less natural motions (e.g., limb twisting). In line with your suggestion, we have provided an anonymized demo link (already submitted to AC following the instructions).
> > > > > > > >
> > > > > > > > 2. **These concerns can lead to difficulties in using the dataset.**
> > > > > > > >
> > > > > > > > As iterated in A(7).3, the dataset is designed to be easily usable for researchers in the multi-person motion prediction field:
> > > > > > > >
> > > > > > > > > “Wang *et al.* [59] aligns the pose representation of four datasets [1, 56, 33, 25] to a unified 15-joint representation, and is followed by other recent works [40, 54, 63]. Therefore, we adopt the same keypoint definition so that future researchers in this field can seamlessly switch between the datasets and easily compare the datasets and methods.”
> > > > > > > >
> > > > > > > > We acknowledge your suggestion that incorporating more information will make this dataset more useful for a broader community. In pursuit of whis, we will 1) include the 3D ball trajectory estimator and visualization tools in our code release as mentioned in [Rebuttal to Reviewer X2WW](https://openreview.net/forum?id=lRu0dN7BY6&noteId=KOXpfZdWpJ) A5; 2) investigate the feasibility of directly estimating parametric human body models like SMPL from multi-view videos to see if we can achieve satisfactory results.
> > > > > > > >
> > > > > > > > 3. **It is not yet clear to me the authors' motivation for choosing action prediction as an example task.**
> > > > > > > >
> > > > > > > > We would like to clarify that social motion prediction is not just an 'example task' auxiliary to the proposed dataset, but rather the primary focus of our work. As outlined at the very beginning of the manuscript:
> > > > > > > >
> > > > > > > > > “Humans exhibit a remarkable capacity for anticipating the actions of others and planning their own actions accordingly. In this study, we strive to replicate this ability by addressing the social motion prediction problem.”
> > > > > > > >
> > > > > > > > Inspired by human cognitive abilities in social motion prediction, we discuss the limitations of existing works and develop our main contributions (introducing a new benchmark, proposing a novel formulation, and designing a cognition-inspired framework for this task). **These points have been extensively discussed in *Introduction* and all reviewers agree that our work is 'well-motivated'.**
> > > > > > > >
> > > > > > > > We agree that motion generation is indeed an important task that could be further explored using our dataset as future work. We will discuss the task and related works in the revised manuscript.

---

> > ### Comment · Reviewer_yiis · 2023-08-13
> > **I read other reviews and comment here**
> >
> > I read other reviews and list my comments to authors, other reviewers, ACs, and SACs.
> >
> > **For Reviewer Nb5i**
> > + For predicting 1s concern, I do not think this is a weakness. Prediction of too-long motions is a generation task exactly. therefore, I suggest authors perform generation results in the experiments, which is a better pretext task.
> >
> > + I have the same concern on annotation, like SMPL(-X), ball, and equality (foot sliding).
> >
> > **For Reviewer 8iac**
> >
> > + For `Lack of survey of related work with respect to the methods`, can these works be compared with your methods as baselines?
> >
> > ---
> >
> > Moreover, I check that the authors clicked the Reproducibility as YES, and I am not sure whether authors should provide (inference) codes in the submission. If not provided, it might be reported to the ethics reviewer.

---

> > > ### Author Response · Authors · 2023-08-21
> > > **Replying to 'I read other reviews and comment here'**
> > >
> > > **Q1: Same concern on annotation, like SMPL(-X), ball, and quality (foot sliding).**
> > >
> > > A1: Please refer to [Rebuttal to Reviewer X2WW](https://openreview.net/forum?id=lRu0dN7BY6&noteId=KOXpfZdWpJ) (A5) for discussions on SMPL(-X) and ball. Please refer to [Rebuttal to Reviewer Nb5i](https://openreview.net/forum?id=lRu0dN7BY6&noteId=356r6ugS16) (A2) for discussions on foot sliding.
> > >
> > > **Q2: Can these works be compared with your methods as baselines?**
> > >
> > > A2: Thank you for your suggestion. We have conducted further comparisons with two recent trajectory prediction methods [a, b] on our dataset. In these comparisons, the models are trained specifically to predict the trajectory, while the local poses of the last input frame are maintained as they are. The results, as shown in the table below, indicate that our method surpasses the performance of these baseline methods.
> > >
> > > |                     | Global   |          |           |           | Local    |          |          |          | Root     |          |          |           |
> > > | ------------------- | -------- | -------- | --------- | --------- | -------- | -------- | -------- | -------- | -------- | -------- | -------- | --------- |
> > > | milliseconds        | 400      | 600      | 800       | 1000      | 400      | 600      | 800      | 1000     | 400      | 600      | 800      | 1000      |
> > > | Social-STGCNN [a] | 188.2    | 246.4    | 298.8     | 346.5     | 62.8     | 82.8     | 98.9     | 112.1    | 173.1    | 225.6    | 273.7    | 317.9     |
> > > | SocialVAE [b]     | 84.0     | 127.2    | 171.7     | 215.1     | 62.8     | 82.8     | 98.9     | 112.1    | 52.6     | 89.7     | 130.3    | 170.8     |
> > > | Ours                | **54.6** | **86.2** | **119.3** | **152.5** | **43.7** | **60.8** | **74.6** | **86.6** | **41.7** | **66.9** | **94.8** | **124.0** |
> > >
> > > **Q3: I am not sure whether authors should provide (inference) codes in the submission.**
> > >
> > > A3: For clarity, you may refer to the guidelines provided in the NeurIPS Paper Checklist: https://neurips.cc/public/guides/PaperChecklist. Under the _Experiments_ section, it is specified that code and data should be submitted if the option is checked. This is distinct from the _Reproducibility_ section. The entire dataset, code, and models, will be publicly released upon acceptance.
> > >
> > >
> > > **References**
> > >
> > > [a] Mohamed, Abduallah, et al. "Social-stgcnn: A social spatio-temporal graph convolutional neural network for human trajectory prediction." CVPR 2020.
> > >
> > > [b] Xu, Pei, Jean-Bernard Hayet, and Ioannis Karamouzas. "Socialvae: Human trajectory prediction using timewise latents." ECCV 2022.

---

> > > > ### Comment · Reviewer_yiis · 2023-08-21
> > > > **Re: Replying to 'I read other reviews and comment here'**
> > > >
> > > > Thanks for the authors' efforts. A2 and A3 resolved my concerns. For the remaining concerns (in `Re: Official Comment by Authors`) and A1 concern, I will discuss them with other reviewers and AC during the following phase.
> > > >
> > > > Best,
> > > >
> > > > Reviewer yiis

---

> ### Comment · Reviewer_yiis · 2023-08-11
> **After rebuttal**
>
> The authors did not submit any rebuttal. Do authors want to participate in author-reviewer discussions?

---

> > ### Author Response · Authors · 2023-08-11
> > **Rebuttal submitted**
> >
> > Dear Reviewer yiis,
> >
> > We appreciate your attention to the review process. We have indeed submitted our 'Author Rebuttal', which includes a general response as well as individual responses to each reviewer's comments. These should be visible to the 'Reviewers Submitted' group. Could you kindly recheck your portal? If there is still an issue, please let us know. To ensure the process continues smoothly, we have also reached out to the conference chairs for their assistance.
> >
> > Thank you for your understanding and cooperation.
> >
> > Best regards,
> >
> > Authors

---

> > > ### Comment · Reviewer_yiis · 2023-08-13
> > >
> > > Dear authors,
> > >
> > > I logged out of the system and refresh the openreview. Nothing changed. I served as a reviewer for 6 submissions this year. This is the only submission that I met bugs. I will keep in touch with PCs. PCs say:
> > > > We will aim to get you to see the other reviews by Monday.
> > >
> > > I will update some parts of my reply ASAP. For parts related to the global rebuttal and the PDF, I will reply to the authors if I see them. Besides, I will also read other reviews and reply more.

---

> > > > ### Author Response · Authors · 2023-08-13
> > > > **Thanks for your kind response**
> > > >
> > > > Dear Reviewer yiis,
> > > >
> > > > Thank you very much for your kind help and understanding in dealing with the ongoing technical issues with the OpenReview system. We understand the inconvenience this situation has created, especially given that this seems to be an isolated issue with our submission.
> > > >
> > > > We greatly appreciate your patience and dedication. We look forward to your updated review and any additional feedback you may provide upon accessing the full review materials.
> > > >
> > > > Best regards,
> > > >
> > > > Authors

---

### Official Review · Reviewer_Nb5i · 2023-07-05

**Soundness:** 2 fair
**Presentation:** 3 good
**Contribution:** 3 good
**Rating:** 4
**Confidence:** 4

**Summary:**

The paper addresses the problem of social motion forecasting utilizing a multi-agent reinforcement learning and combines behavioral cloning and generative adversarial imitation learning. Social and strategic interactions are modeled in a “cognitive hierarchy framework”. The paper further introduces a new large-scale 3D multi-human motion dataset based on no-dribble-3-on-2 basketball.

**Strengths:**

The paper is, in general, well-written and the method well-motivated. The novel dataset is highly relevant in the under-explored area of multiple human motion forecasting.

**Weaknesses:**

In general, 1s of forecast for such complex social interactions is too short, as players mostly just continue their motion, e.g. swing of an arm. In basketball there is the 5s rule, I would thus suggest to at least forecast 5s of future motion to get a good understanding of the interactions and of the game. Instead of MPJPE metrics for long-term motion forecasting could be employed, e.g. [b,c].

The method results in the supplementary video exhibit a lot of foot sliding, suggesting a generalization problem for walking motions. I would assume that this is due to using 3D velocities as actions, which are translation invariant but crucially not rotation invariant. The choice of a rotation-variant actions is surprising as the introduced dataset (almost) plays out on a circle, with 3 players on the circle and 2 players within. Did the authors try to alleviate the issue by augmenting the data?

The authors say that "Human motions exhibit greater dynamism in terms of pose diversity and movement speed, making motion prediction more challenging than in previous datasets” (LL42-44) but then say that players are not allowed to “dribble” (L111 & L115) - does that not significantly reduce the range of motion, i.e. there is little global translation?

The authors should elaborate how they obtained body joint angles (L144), Figure 2. In Figure 1 it seems that the authors extract SMPL parameters as pose representation but it is never specified - however, extracting angular representations from 3D joint locations is non-trivial and the authors need to explain how those are produced. The lack of this information makes Figure 2 unconvincing: assuming SMPL was used, the skeletal structure is quite different from the parametric model used in CMU [1], we thus would expect significant differences in angular distributions. For example, hinge joints in CMU are zero everywhere except at the hinge dimension - is this the case in the authors representation as well?

The paper should replicate the MRT [59] experiments - at least for the available datasets CMU Mocap and MuPoTS-3D.

HRI [32] originally was trained on local (normalized) motion and not on global coordinates: in this work, are the global coordinates passed to HRI for training and evaluation? How would HRI perform with normalization such as subtracting the root location at the last input frame? Such normalization can be undone trivially and should greatly improve the (short-term) motion prediction results. The evaluation as is is unfair for HRI if global coordinates are passed.


**Minor issues**

LL54-55: While CHT [9] is cited and put into context (LL98-105) a short overview would be helpful, as it is not clear from the paper what the authors mean by “k represents the depth of strategic thought” (L55).

An interesting dataset for evaluating the approach would be the Haggling dataset [a] where persons play a triadic haggling game, an adversarial game between two players to sell a product to a third person (buyer). This work should be at least cited and compared against in Table 1.

In the context of 3D human motion forecasting planning or “intention” has been addressed as intermediate signal in previous works [c,d] and should be cited.

[a] Joo, Hanbyul, et al. "Towards social artificial intelligence: Nonverbal social signal prediction in a triadic interaction." CVPR 2019.

[b] Gopalakrishnan, Anand, et al. "A neural temporal model for human motion prediction." Proceedings of the IEEE/CVF Conference on Computer Vision and Pattern Recognition. 2019.

[c] Tanke, Julian, Chintan Zaveri, and Juergen Gall. "Intention-based long-term human motion anticipation." 3DV 2021.

[d] Diller, Christian, Thomas Funkhouser, and Angela Dai. "Forecasting characteristic 3D poses of human actions." Proceedings of the CVPR 2022.

**Rebuttal**
I appreciate the effort made by the authors to answer my questions.

However, I still believe that 1s is too short for what the authors claim their dataset is capable of demonstrating: intricate "strategic" social interplay.

**Questions:**

I wonder if it would be possible to evaluate the combined motion of all persons, e.g. not just of the per-person poses but also the global configuration of the poses is sensible or not. Imagine, for example, one person throwing the ball to their left-hand person but the right-hand person generates the "receive" motion - which would be unlikely.

It is unclear which version of PoseTrack the authors address in Table 1: PoseTrack17 [23] contains 556 sequences (and not 13) and PoseTrack18 [5] contains 1138 sequences. Could the authors elaborate if this version is a subset of PoseTrack?

I would assume this is an encoding error but in the supplementary video every 4 to 5th frame “freezes” making it a bit more difficult to check the results frame by frame in video (QuickTime Player, OSX).

What was the motivation of color choice for Visual representation of the Cognitive Hierarchy Visualization: I would suggest to use warm colors for the offense and cold colors for the defense as especially purple and rose are too close to each other.

**Limitations:**

The authors address some limitations, notably the assumption that a certain cognitive hierarchy is present in the data.
It would be interesting to see how far into the future the model can predict without producing failing poses or freezing up.

---

> ### Author Rebuttal · Authors · 2023-08-10
>
> We sincerely appreciate the reviewer’s constructive and detailed feedback. We are grateful that the you find our paper 'well-written', our proposed 'well-motivated', and our dataset 'highly relevant in the under-explored area'. In the following, we aim to address your concerns:
>
> **Q1: 1s is too short.**
>
> A1: We appreciate the reviewer's insightful suggestions. We understand the importance of longer-term predictions in understanding complex social interactions and the dynamics of the game. However, we would like to provide the following points to clarify our choice of a 1-second forecast horizon:
>
> - Basketball is characterized by its swift pace, whereby a significant amount of action can occur within a 1-second timeframe. In full-court matches, possession often concludes within few seconds. Our research scenario is even more intense and compact compared to conventional basketball games, and it operates at a faster pace.
>
> - Our video demonstration reveals that the 1-second future doesn't merely continue past motions. Instead, it involves defensive switches and sudden changes in moving directions. As a matter of fact, our level-1 policy is primarily motion continuation, but our policy network learns complex strategic behaviors at higher levels through the proposed cognitive hierarchy design.
>
> - We opted not to use a longer prediction horizon primarily because future motions heavily depend on the uncertain outcomes of passes. For instance, if player A is passing to player B, the subsequent motions will be greatly influenced by whether the ball is successfully received by B or stolen midway. Predicting such outcomes based on current information poses a considerable challenge for any model.
>
> We plan to extend our datasets to include regular basketball games with more information and explore longer prediction lengths for future research.
>
>
> **Q2: Foot sliding.**
>
> A2: Foot sliding might be partly due to our model not explicitly regularizing foot contact velocities as previous works have done [R17, R18, R19]. Our research primarily targets predicting the strategic action decisions rather than the quality of the motion itself. Hence, we did not initially design our model to address these detailed aspects of the motion quality. Based on the reviewer's suggestion, we plan to explore training techniques such as foot contact loss and data augmentation to enhance the visual quality of the motion predictions. We believe these techniques could potentially alleviate the foot sliding issue.
>
>
> **Q3: Little global translation?**
>
> A3: The no-dribbling rule in this practice enforces the players to make more strategic decisions. While dribbling is prohibited for the offensive players, it doesn't necessarily result in a significant decrease in the overall range of motion. It's important to note that defensive players are not under the same restriction. They are required to continually move back and forth to maintain adequate defense coverage. This requirement leads to substantial global translation, as evidenced by the video we have provided. Therefore, while the scope of certain specific motions is reduced, the overall diversity and dynamism of human motions are preserved, and the challenge of predicting those motions remains high. The large root error (especially for the 'frozen' baseline) in Table 2 of the submission also demonstrate this idea.
>
> **Q4: Body joint angles.**
>
> A4: As discussed in General Response Q2, our dataset use 3D pose represetation. Previous work MRT [59] converts the CMU-Mocap dataset to a format of 15-joint 3D poses. We strictly followed its definition and perform data analysis on the aligned skeletal structure. We directly computed the angle of intersection formed by the hinged limbs since we do not have the limb twists of the 3D pose representations.
>
> **Q5: Replicate the MRT experiments.**
>
> A5: Following your suggestion, we train and test our method on the CMU-Mocap (UMPM) dataset [40]. Please refer to Table 1 of the rebuttal PDF for the results. Notably, our method exhibits particular strength in predicting long-term root trajectories and global motions.
>
> **Q6: Details on HRI.**
>
> A6: When training and evaluating HRI, we pass in normalized motions following the original settings. Therefore, it achieves respectable short-term motion prediction accuracy in Table 2, as supposed by the reviewer.
>
> **Q7: CHT overview.**
>
> A7: We provide an intuitive explanation on CHT and level-k reasoning in Line 51-55 and Line 195-197. We will add a short overview to Section 2.3 following your suggestion.
>
> **Q8: Haggling dataset.**
>
> A8: Please refer to General Response Q1. We will cite and compare with it following your suggestion.
>
> **Q9: Intention as intermediate signal.**
>
> A9: Thank you for the information, we will add them to Section 2.
>
> **Q10: Evaluating the combined motion.**
>
> A10: The GAIL global discriminator $D$ is developed on a similar idea, which penalizes global state-action pairs that are unlikely to occur. For a quantitative evaluation of the combined motion of all persons, incorporating a user study could be beneficial. We appreciate this suggestion and plan to include such a study in our future work.
>
> **Q11: PoseTrack version.**
>
> A11: For PoseTrack, we compare with the version provided by SoMoF [21], which may cause confusion in number of sequences given different definitions. We apologize for the confusion. We will change the dataset name to 'PoseTrack (SoMoF)' and revise the sequence numbers in next version.
>
> **Q12: Video encoding error.**
>
> A12: Thank you for bringing this issue to our attention. We have thoroughly reviewed the supplementary video and were unable to replicate the issue. Could you please provide us with your OS and QuickTime Player version? We will ensure that our video encoding is widely compatible in the future release.
>
> **Q13: Color choice.**
>
> A13: Thanks for your suggestion! We will change the visualization colors accordingly.

---

> > ### Author Response · Authors · 2023-08-11
> > **References (also appended in the General Response)**
> >
> > [R17] Tevet, Guy, et al. "Human motion diffusion model." ICLR 2023.
> >
> > [R18] Tseng, Jonathan, Rodrigo Castellon, and Karen Liu. "Edge: Editable dance generation from music." CVPR 2023.
> >
> > [R19] Rempe, Davis, et al. "Humor: 3d human motion model for robust pose estimation." ICCV 2021.

---

### Official Review · Reviewer_8iac · 2023-07-06

**Soundness:** 3 good
**Presentation:** 4 excellent
**Contribution:** 2 fair
**Rating:** 5
**Confidence:** 4

**Summary:**

The paper has two major contributions to the field of multi-person motion detection:

1) A new open source dataset that is sufficiently large scale as compared to those already available. But more importantly is dense in terms of strategic interactions and more diversity of action distributions for reinforcement learning algorithms. This is very relevant to the field.

2) To demonstrate the efficacy of the datasets, they also propose a new initiation learning framework combining behavior cloning as well as  GAIL.

**Strengths:**

1) The paper is very well written with clear figures to make the concepts easy to understand.
2) The dataset is a welcome addition to the space of close space multi agent interaction modeling.
3) The experimental section is very detailed with an ablative section to help understand the implications of each of the contributions of the MARL framework.

**Weaknesses:**

The paper suffers from two major issues. The authors ignore a large body of work that are not directly related to the close space interactions but highly relevant to the field of multi-agent trajectory prediction. This includes both datasets as well as methods available to get state of the art results in the domain.

1) Lack of survey of related work with respect to the dataset: Many work in the autonomous driving domain feature highly complex social scenarios with many agents interacting with one other. The datasets include industrial benchmarks like Argoverse, nuScenes, Waymo Open Motion Dataset [1] amongst others. While these datasets including not just pedestrian trajectories but also other road users like vehicles and cyclists, a comparison must be made about how the complex multi-agent crowd level interactions are different and similar to the close space interaction modeling that is being studied in this paper. The crowds of pedestrian in these scene perform complex social maneuvers of working cooperatively ( cross walks) and destructively (jaywalking) as examples and how these strategies are relevant to this field is an important comparison. Given the existence of these other much larger dataset in terms of number of subjects available, it will be important to understand what benefits the new dataset adds to the field.

2) Lack of survey of related work with respect to the methods: On the datasets mentioned above a large body of work exists involving both simple deep learning transformer and non transformer based Bayesian methods as well as reinforcement learning frameworks [4] (as an example) that the MARL framework needs to be compared against.


[1] Waymo Open Motion Dataset

[2] Symphony: Learning Realistic and Diverse Agents for Autonomous Driving Simulation

**Questions:**

Would it be possible to include the relevant recent work in the field of multi-agent human trajectory prediction?

---

> ### Author Rebuttal · Authors · 2023-08-10
>
> We sincerely appreciate the reviewer’s constructive and positive feedback. We are happy that the you find our paper 'well-written', our proposed dataset 'a welcome addition', and our experiments 'very detailed'. In the following, we aim to address your questions and concerns:
>
> **Q1: Lack of survey of related work with respect to the dataset.**
>
> A1: Thank you very much for the suggestion! We provide a discussion on the multi-agent trajectory prediction datasets in General Response Q1. We will cite and discuss these works in Section 2.1 in the next revision following your suggestion.
>
> **Q2: Lack of survey of related work with respect to the methods.**
>
> A2: Thank you very much for the suggestion! We referred to the relevant recent work in the field of multi-agent human trajectory prediction in Line 20-26. We will cite and discuss more research works in this area in Section 2.2 in the next revision following your suggestion. By the way, is part of the review references ([4]) missing?

---

### Official Review · Reviewer_X2WW · 2023-07-07

**Soundness:** 3 good
**Presentation:** 2 fair
**Contribution:** 3 good
**Rating:** 5
**Confidence:** 4

**Summary:**

This paper aims to predict multiperson human motion. The main contributions of this paper are:
1. The paper presents a large-scale multi-human 3D motion dataset with intense, strategic interactions.
2. The paper formulates the multiperson human prediction problem as MARL and presents a hierarchy framework to model complex social interactions.


**Strengths:**

The paper has several strengths:
1. Overall, the paper is well-written with a clear and well-motivated introduction. The hierarchy framework is introduced to consider the recursive decisions in the interaction.
2. The paper proposed a large-scale multi-person motion dataset
3. Evaluation on the proposed dataset demonstrates that the method outperforms state-of-the-art methods.


**Weaknesses:**

1. In L32-34, the paper claims that most existing methods employ end-to-end supervised training which overlooks the cognitive aspects. Considering the state (past motion) and action (future velocity) space defined in this paper, could the author provide more insight on the difference between the BC, GAIL used in the paper and GAN-based supervised learning in related work?
2. While this work qualitatively demonstrates the utility of the proposed framework for tasks specific to motions like sports, it would be beneficial to provide additional comparisons on existing benchmarks in addition to Figure 4.
3. Other multi-person datasets in sports also exist, such as KTH Multiview Football datasets. The paper may compare the proposed dataset and method with them.



**Questions:**

Please check 1. - 3. in Weaknesses.
In addition, why was the best performance achieved when K=3? if the claims of the cognitive hierarchy are true, I was wondering what happened after having more levels of decision.



**Limitations:**

The visualization of the output from each level shows some existence of cognitive hierarchies, but it should be more clear if the paper and the demo video could include the visuals in meshes (with ball movement if possible) like Figure 1.
Overall, the author's response to the concerns in the Weakness section is needed to make the final decision. I am happy to increase the rating if my concerns are addressed.

---

> ### Author Rebuttal · Authors · 2023-08-10
>
> We sincerely appreciate the reviewer's constructive and positive feedback. We are grateful that you recognize our main contributions and find it 'well-written with a clear and well-motivated introduction'. In the following, we aim to address your questions and concerns:
>
> **Q1: Provide more insight on the difference between the BC, GAIL and GAN-based supervised learning.**
>
> A1: Wang _et al._ [57] initially introduces an imitation learning formulation for single-person human motion prediction with BC and GAIL compared to end-to-end supervised training. Our work extends conceptually from [24, 57] as it adapts BC and GAIL for a multi-agent imitation learning context. Our approach is distinct from GAN-based supervised learning in two primary ways:
>
> - The GAIL objective focuses on making the state-action pairs produced by the joint policies of agents and that of experts indistinguishable. This is in contrast to the GAN framework (_e.g._, [59]), where the objective is to make the predicted future motion indistinguishable from the real ones.
>
> - Our MARL formulation enables a more principled integration with the cognitive hierarchy theory, while existing works overlook the cognitive aspects of human social action planning. By modeling the motion prediction process with chained policy networks, we derive the BC and GAIL regularization for each level accordingly, and propose to share parameters for policy networks $\phi_{(1)} \dots \phi_{(K)}$, both of which lead to performance improvements as shown in Table 3 of the submitted manuscript.
>
> **Q2: It would be beneficial to provide additional comparisons on existing benchmarks.**
>
> A2: Following your suggestion, we train and test our method on the CMU-Mocap (UMPM) dataset [40]. Please refer to Table 1 of the rebuttal PDF for the results. Our approach outperforms the previous methods [32, R2, 59] and is comparable with a concurrent work [40]. Notably, our method exhibits particular strength in predicting long-term root trajectories and global motions.
>
> **Q3: Other multi-person datasets in sports also exist, such as KTH Multiview Football datasets.**
>
> A3: Please refer to General Response Q1 for the discussions. We will make it more clear in the revision.
>
> **Q4: Why was the best performance achieved when K=3?**
>
> A4: Our cognition-inspired framework aims to learn joint policies from real-world expert demonstrations and to discover the explainable cognitive hierarchies. The finding that $K=3$ provides the best fit for the real-world dataset may suggest that this particular depth of strategic reasoning most aptly encapsulates the human decision-making process as captured in our collected data. As we increase the number of hierarchical levels ($K>3$), we continue to observe the cognitive hierarchy visualizations among different levels, though there is a slight increase in the mean prediction error.
>
>
> **Q5: Could include the visuals in meshes (with ball movement if possible).**
>
> A5: Thanks for your suggestion!
>
>   - **Body Mesh**: As discussed in General Response Q2, we use 3D poses in the experiments. While the 3D poses can certainly be fitted to a parametric body mesh (such as SMPL, as shown in Figure 1) for a more intuitive visualization, this process may introduce additional errors. To avoid these potential complications, we chose to present the 3D skeletons as they are -- the direct output of the model, free from potential distortions introduced by fitting. This approach makes it easier to identify potential issues. However, in response to your suggestion, we will include the mesh fitting and visualization scripts in our code release.
>
>   - **Ball movement**: We implemented a 3D ball trajectory estimator using an object detector and multi-view triangulation. However, we chose not to include the ball information in model training for two reasons: 1) to ensure a fair comparison with previous methodologies, and 2) due to the fact that ball moves significantly faster than humans, which sometimes result in inaccurate estimations caused by motion blur and detection failures. We are committed to improving our ball detection algorithms, and in response to your suggestion, we will include the 3D ball trajectory estimator and visualization tools in our code release.

---

### Official Review · Reviewer_EKS7 · 2023-07-13

**Soundness:** 3 good
**Presentation:** 3 good
**Contribution:** 3 good
**Rating:** 4
**Confidence:** 4

**Summary:**

--

**Strengths:**

--

**Weaknesses:**

--

**Questions:**

--

**Limitations:**

--

---

> ### Author Rebuttal · Authors · 2023-08-10
>
> Thank you for taking the time to provide your feedback. However, we noticed that the specific details of your review were not included. To better understand your viewpoint and address any potential issues, it would be immensely helpful if you could provide more detailed feedback.

---

### Author Rebuttal · Authors · 2023-08-10

# General Response

We sincerely thank all reviewers for their meticulous reviews and constructive remarks. It is heartening to note that all reviewers have acknowledged the motivation behind our work, as well as its main contributions — especially the proposed method. We will incorporate discussion of relevant works and additional experiments in the ﬁnal version accordingly. In the following, we will first address some shared issues on the proposed dataset, then respond to each reviewer's comments respectively.

**Q1: Dataset comparisons**

A1: We discuss the existing multi-person motion datasets in Section 2.1 and provide a comparison with some of them (Table 1 of the submited manuscript). The table contains comparisons with _'existing multi-person motion datasets employed by previous works on the multi-person motion prediction task'_ (table caption). In other words, we primarily compare with the utilized datasets within this domain as referenced in previous works [2, 3, 58, 59, 19, 54, 40, 63]. We recognize that there are other multi-person datasets that are relevant to our work, as recommended by the reviewers. Consequently, we present a brief discussion on the differences between our work and each of these datasets below. **These works will also be cited and discussed in Section 2.1 and Table 1 of the revised manuscript following your suggestions.**



- **Multi-person Sports Datasets (Reviewer X2WW).** There exists a variety of multi-person sports datasets for group behavior understanding, e.g. NBA [R3], MultiSports [R4], Volleyball [R5], NCAA [R6]. However, these datasets are primarily designed for action recognition from RGB videos and lack pose annotations or 3D information. Felsen _et al._ [R7] collect a water polo dataset and a basketball dataset for group trajectory prediction, but they only contain trajectory-level data rather than human poses. The KTH Multiview Football dataset [R8] consists of two subsets (2D and 3D), with the 3D subset containing merely 800 time frames (32 seconds) for two isolated players. Therefore, it might be more suitable for testing or few-shot adaptation.
- **Multi-agent trajectory prediction datasets in autonomous driving (Reviewer 8iac).** As suggested by the reviewer, there are a large body of work on multi-agent trajectory prediction that is _'not directly related to'_ our task, but _'highly relevant'_. We briefly discussed this research area in Line 20-26. Given the diffrent task objectives, the datasets also exhibit different characteristics. The trajectory datasets [R9, R10, R11, R12] possess advanteges such as larger scale and diverse classes of agents. Nevertheless, these datasets only contain coarse-grained interactions at the trajectory level (mainly collision avoidance) and fail to capture the rich and fine-grained human actions. Furthermore, our dataset features intense and strategic interactions, such as frequent sudden direction changes and deception maneuvers, which are largely absent from existing datasets. Thus, our dataset can serve as a valuable supplement to the existing ones, posing additional challenges to this research field.


- **Haggling dataset [R13] (Reviewer Nb5i)**. The Haggling dataset shares relevance with our work as that it also invloves social interactions and body gesture prediction. The main difference is that they focus on a triadic conversational scenario, therefore the interactions are relatively simpler, and the body motions are more static.

- **UMPM benchmark [R1] (Reviewer yiis).** The UMPM dataset is a multi-person mocap dataset. As per the paper, 2/9 of the scenarios contain interaction between the subjects and is suitable for our task: _(6) a conversation with natural gestures, and (7) the subjects throw or pass a ball to each other while walking around_. Compared to it, our dataset is advantageous in terms of a larger number of subjects and more strategic interactions. We also conduct experiments on an extended version of UMPM following [40] and the results are reported in Table 1 of the rebuttal PDF.

- **InterGen [R14] (Reviewer yiis).** This concurrent work introduces a multimodal dataset called InterHuman for text-to-multi-human motion generation. Consequently, their primary objective is to cover a diverse range of interactions and annotate with extensive textual descriptions. In contrast, the focus of our dataset is intense and strategic human interactions for social motion prediction. We believe that both our dataset and InterHuman offer valuable additions to the research community, albeit with different task focuses.

**Q2: Data format**

A2: Our work primarily focuses on 3D body keypoint sequences following previous studies [59, 19, 54, 57, 40, 32], as discussed in supplementary material Line 26-28. The 3D skeletons are also the standard product of markerless multi-view motion capture methods [64, R15, R16] which are more suitable for our data collection scenario than marker-based ones. We use $J=15$ skeleton joints with the same difinition with previous works [40, 59].

---

## Response to Ethics Reviews

We thank the Ethics Reviewers for taking the time to review our paper and detailed suggestions.

1. The athletes were indeed informed that their identities would be kept private. We obtained written consent from each participant, clearly outlining the data collection process and intended uses, ensuring transparency. We will include the consent forms given to participants in the Appendix in the revisions.

2. While gait has been acknowledged as a behavioral biometric in recent studies, gait recognition primarily relies on video or sensor data. It is crucial to clarify that our released dataset does not contain video or sensor statistics, mitigating the risks associated with biometric identification. To our best knowledge, there is no viable solution to recognize person identities from the data we provided given no extra information.

---

> ### Author Response · Authors · 2023-08-11
> **References (also appended to the PDF)**
>
> [R1] Van der Aa, N. P., et al. "Umpm benchmark: A multi-person dataset with synchronized video and motion capture data for evaluation of articulated human motion and interaction." ICCVW, 2011.
>
> [R2] Dang, Lingwei, et al. "Msr-gcn: Multi-scale residual graph convolution networks for human motion prediction." ICCV 2021.
>
> [R3] Yan, Rui, et al. "Social adaptive module for weakly-supervised group activity recognition." ECCV 2020.
>
> [R4] Li, Yixuan, et al. "Multisports: A multi-person video dataset of spatio-temporally localized sports actions." ICCV 2021.
>
> [R5] Ibrahim, Mostafa S., et al. "A hierarchical deep temporal model for group activity recognition." CVPR 2016.
>
> [R6] Ramanathan, Vignesh, et al. "Detecting events and key actors in multi-person videos." CVPR 2016.
>
> [R7] Felsen, Panna, Pulkit Agrawal, and Jitendra Malik. "What will happen next? forecasting player moves in sports videos." ICCV 2017.
>
> [R8] Kazemi, Vahid, et al. "Multi-view body part recognition with random forests." BMVC 2013.
>
> [R9] Ettinger, Scott, et al. "Large scale interactive motion forecasting for autonomous driving: The waymo open motion dataset." ICCV 2021.
>
> [R10] Igl, Maximilian, et al. "Symphony: Learning realistic and diverse agents for autonomous driving simulation." 2022 International Conference on Robotics and Automation (ICRA). IEEE, 2022.
>
> [R11] Chang, Ming-Fang, et al. "Argoverse: 3d tracking and forecasting with rich maps." Proceedings of the IEEE/CVF conference on computer vision and pattern recognition. 2019.
>
> [R12] Caesar, Holger, et al. "nuscenes: A multimodal dataset for autonomous driving." Proceedings of the IEEE/CVF conference on computer vision and pattern recognition. 2020.
>
> [R13] Joo, Hanbyul, et al. "Towards social artificial intelligence: Nonverbal social signal prediction in a triadic interaction." CVPR 2019.
>
> [R14] Liang, Han, et al. "InterGen: Diffusion-based Multi-human Motion Generation under Complex Interactions." arXiv preprint arXiv:2304.05684 (2023).
>
> [R15] Hanyue Tu, Chunyu Wang, and Wenjun Zeng. "Voxelpose: Towards multi-camera 3d human pose estimation in wild environment." ECCV 2020.
>
> [R16] Jianfeng Zhang, Yujun Cai, Shuicheng Yan, and Jiashi Feng. "Direct multi-view multi-person 3d pose estimation." NeurIPS 2021.
>
> [R17] Tevet, Guy, et al. "Human motion diffusion model." ICLR 2023.
>
> [R18] Tseng, Jonathan, Rodrigo Castellon, and Karen Liu. "Edge: Editable dance generation from music." CVPR 2023.
>
> [R19] Rempe, Davis, et al. "Humor: 3d human motion model for robust pose estimation." ICCV 2021.

---

### Author Response · Authors · 2023-08-13
**[EMERGENCY] Reviewers can not see the rebuttal.**

Dear PCs, ACs, and reviewers,

We are writing regarding a potential issue with the visibility of our submitted 'Author Rebuttal' in OpenReview (Submission Number: 534). **One of the reviewers, Reviewer yiis, has mentioned in a recent communication that they were unable to see our rebuttal.**

**We have confirmed on our end that the 'Author Rebuttal', including a general response as well as individual responses to each reviewer's comments, has indeed been submitted and should be visible to the 'Reviewers Submitted' group.** We kindly ask for your assistance in resolving this matter.

It is essential for us to ensure that all reviewers have access to our responses, as we believe the rebuttal provides important context and clarification to our paper. We understand that these are technical issues that can occasionally arise and we appreciate your support in addressing this.

Please do not hesitate to contact us if you need any further information from our end.

Thank you for your attention and assistance in this matter.

Best regards,

Paper 534 Authors

---

> ### Author Response · Authors · 2023-08-14
> **[UPDATE] Reposting the rebuttals due to visibility issues**
>
> We have just reposted our rebuttal, including the general and individual responses to each reviewer's comments, using the _Official Comment_ tool. Our goal is to guarantee that all reviewers have access to our responses and can engage in any further discussions before the deadline.
>
> We apologize for any inconvenience this may have caused and express our appreciation for your patience.

---

> > ### Comment · Reviewer_yiis · 2023-08-15
> >
> > I can see all information. You don’t have to repost it.

---

### Author Response · Authors · 2023-08-14
**General Response (Repost)**

We sincerely thank all reviewers for their meticulous reviews and constructive remarks. It is heartening to note that all reviewers have acknowledged the motivation behind our work, as well as its main contributions — especially the proposed method. We will incorporate a discussion of relevant works and additional experiments in the ﬁnal version accordingly. In the following, we will first address some shared issues on the proposed dataset, then respond to each reviewer's comments respectively.

**Q1: Dataset comparisons**

A1: We discuss the existing multi-person motion datasets in Section 2.1 and provide a comparison with some of them (Table 1 of the submitted manuscript). The table contains comparisons with _'existing multi-person motion datasets employed by previous works on the multi-person motion prediction task'_ (table caption). In other words, we primarily compare with the utilized datasets within this domain as referenced in previous works [2, 3, 58, 59, 19, 54, 40, 63]. We recognize that there are other multi-person datasets that are relevant to our work, as recommended by the reviewers. Consequently, we present a brief discussion on the differences between our work and each of these datasets below. **These works will also be cited and discussed in Section 2.1 and Table 1 of the revised manuscript following your suggestions.**

- **Multi-person Sports Datasets (X2WW).** There exists a variety of multi-person sports datasets for group behavior understanding, e.g. NBA [R3], MultiSports [R4], Volleyball [R5], NCAA [R6]. However, these datasets are primarily designed for action recognition from RGB videos and lack pose annotations or 3D information. Felsen _et al._ [R7] collect a water polo dataset and a basketball dataset for group trajectory prediction, but they only contain trajectory-level data rather than human poses. The KTH Multiview Football dataset [R8] consists of two subsets (2D and 3D), with the 3D subset containing merely 800 frames (32 seconds) for two isolated players. Therefore, it might be more suitable for testing or few-shot adaptation.
- **Multi-agent trajectory prediction datasets in autonomous driving (Reviewer 8iac).** As suggested by the reviewer, there is a large body of work on multi-agent trajectory prediction that is _'not directly related to'_ our task, but _'highly relevant'_. We briefly discussed this research area in Line 20-26. Given the different task objectives, the datasets also exhibit different characteristics. The trajectory datasets [R9, R10, R11, R12] possess advantages such as larger scale and diverse classes of agents. Nevertheless, these datasets only contain coarse-grained interactions at the trajectory level (mainly collision avoidance) and fail to capture the rich and fine-grained human actions. Furthermore, our dataset features intense and strategic interactions, such as frequent sudden direction changes and deception maneuvers, which are largely absent from existing datasets. Thus, our dataset can serve as a valuable supplement to the existing ones, posing additional challenges to this research field.

- **Haggling dataset [R13] (Nb5i)**. The Haggling dataset shares relevance with our work as that it also invloves social interactions and body gesture prediction. The main difference is that they focus on a triadic conversational scenario, therefore the interactions are relatively simpler, and the body motions are more static.

- **UMPM benchmark [R1] (yiis).** The UMPM dataset is a multi-person mocap dataset. As per the paper, 2/9 of the scenarios contain interaction between the subjects and is suitable for our task: _(6) a conversation with natural gestures, and (7) the subjects throw or pass a ball to each other while walking around_. Compared to it, our dataset is advantageous in terms of a larger number of subjects and more strategic interactions. We also conduct experiments on an extended version of UMPM following [40] and the results are reported in Table 1 of the rebuttal PDF.

- **InterGen [R14] (yiis).** This concurrent work introduces a multimodal dataset called InterHuman for text-to-multi-human motion generation. Consequently, their primary objective is to cover a diverse range of interactions and annotate with extensive textual descriptions. In contrast, the focus of our dataset is intense and strategic human interactions for social motion prediction. We believe that both our dataset and InterHuman offer valuable additions to the research community, albeit with different task focuses.

**Q2: Data format**

A2: Our work primarily focuses on 3D body keypoint sequences following previous studies [59, 19, 54, 57, 40, 32], as discussed in supplementary material Line 26-28. The 3D skeletons are also the standard product of markerless multi-view motion capture methods [64, R15, R16] which are more suitable for our data collection scenario than marker-based ones. We use $J=15$ skeleton joints with the same definition as previous works [40, 59].

---

> ### Author Response · Authors · 2023-08-14
> **References**
>
> [R1] Van der Aa, N. P., et al. "Umpm benchmark: A multi-person dataset with synchronized video and motion capture data for evaluation of articulated human motion and interaction." ICCVW, 2011.
>
> [R2] Dang, Lingwei, et al. "Msr-gcn: Multi-scale residual graph convolution networks for human motion prediction." ICCV 2021.
>
> [R3] Yan, Rui, et al. "Social adaptive module for weakly-supervised group activity recognition." ECCV 2020.
>
> [R4] Li, Yixuan, et al. "Multisports: A multi-person video dataset of spatio-temporally localized sports actions." ICCV 2021.
>
> [R5] Ibrahim, Mostafa S., et al. "A hierarchical deep temporal model for group activity recognition." CVPR 2016.
>
> [R6] Ramanathan, Vignesh, et al. "Detecting events and key actors in multi-person videos." CVPR 2016.
>
> [R7] Felsen, Panna, Pulkit Agrawal, and Jitendra Malik. "What will happen next? forecasting player moves in sports videos." ICCV 2017.
>
> [R8] Kazemi, Vahid, et al. "Multi-view body part recognition with random forests." BMVC 2013.
>
> [R9] Ettinger, Scott, et al. "Large scale interactive motion forecasting for autonomous driving: The waymo open motion dataset." ICCV 2021.
>
> [R10] Igl, Maximilian, et al. "Symphony: Learning realistic and diverse agents for autonomous driving simulation." 2022 International Conference on Robotics and Automation (ICRA). IEEE, 2022.
>
> [R11] Chang, Ming-Fang, et al. "Argoverse: 3d tracking and forecasting with rich maps." Proceedings of the IEEE/CVF conference on computer vision and pattern recognition. 2019.
>
> [R12] Caesar, Holger, et al. "nuscenes: A multimodal dataset for autonomous driving." Proceedings of the IEEE/CVF conference on computer vision and pattern recognition. 2020.
>
> [R13] Joo, Hanbyul, et al. "Towards social artificial intelligence: Nonverbal social signal prediction in a triadic interaction." CVPR 2019.
>
> [R14] Liang, Han, et al. "InterGen: Diffusion-based Multi-human Motion Generation under Complex Interactions." arXiv preprint arXiv:2304.05684 (2023).
>
> [R15] Hanyue Tu, Chunyu Wang, and Wenjun Zeng. "Voxelpose: Towards multi-camera 3d human pose estimation in wild environment." ECCV 2020.
>
> [R16] Jianfeng Zhang, Yujun Cai, Shuicheng Yan, and Jiashi Feng. "Direct multi-view multi-person 3d pose estimation." NeurIPS 2021.
>
> [R17] Tevet, Guy, et al. "Human motion diffusion model." ICLR 2023.
>
> [R18] Tseng, Jonathan, Rodrigo Castellon, and Karen Liu. "Edge: Editable dance generation from music." CVPR 2023.
>
> [R19] Rempe, Davis, et al. "Humor: 3d human motion model for robust pose estimation." ICCV 2021.

---

> ### Author Response · Authors · 2023-08-14
> **Tables in the Rebuttal PDF (Repost)**
>
> Table 1: Comparison with state-of-the-art methods on the CMU-Mocap (UMPM) dataset [40], which merges UMPM [R1] with CMU-Mocap [1]. Our approach outperforms the previous methods [32, R2, 59] and is comparable with a concurrent work [40].
>
> |                | Global |         |         |         | Local  |        |         |        | Root   |        |         |        |
> | -------------- | ------ | ------- | ------- | ------- | ------ | ------ | ------- | ------ | ------ | ------ | ------- | ------ |
> | milliseconds   | 200    | 600     | 1000    | Avg.    | 200    | 600    | 1000    | Avg.   | 200    | 600    | 1000    | Avg.   |
> | HRI [32]       | 49     | 130     | 207     | 129     | 41     | 97     | 130     | 89     | 31     | 90     | 158     | 93     |
> | MSR [R2]       | 53     | 146     | 231     | 143     | 46     | 106    | 137     | 96     | 29     | 94     | 175     | 99     |
> | MRT [59]       | 36     | 115     | 192     | 114     | 36     | 108    | 159     | 101    | 27     | 88     | 157     | 91     |
> | TBIFormer [40] | **30** | **109** | 182     | 107     | **27** | **84** | **118** | **76** | **18** | 72     | 133     | 74     |
> | Ours           | 35     | 106     | **173** | **105** | 33     | 89     | 122     | 81     | 20     | **67** | **122** | **70** |
>
> ---
>
> Table 2: Ablation study on weight $\lambda$. We compute the mean prediction errors (mm) of global pose, local pose, and root, respectively. We use $\lambda=0.002$ in the main manuscript.
>
> |                  | Global   |          |           |           | Local    |          |          |          | Root     |          |          |           |
> | ---------------- | -------- | -------- | --------- | --------- | -------- | -------- | -------- | -------- | -------- | -------- | -------- | --------- |
> | milliseconds     | 400      | 600      | 800       | 1000      | 400      | 600      | 800      | 1000     | 400      | 600      | 800      | 1000      |
> | $\lambda=0.01$   | 61.3     | 94.1     | 127.5     | 160.2     | 50.3     | 69.3     | 84.9     | 97.4     | 45.6     | 71.1     | 99.2     | 128.0     |
> | $\lambda=0.005$  | 54.8     | 86.9     | 120.5     | 154.6     | 43.8     | 61.1     | 74.9     | 87.4     | 41.8     | 67.4     | 97.6     | 126.2     |
> | $\lambda=0.002$  | 54.6     | 86.2     | **119.3** | **152.5** | 43.7     | 60.8     | 74.6     | 86.6     | 41.7     | 66.9     | **94.8** | **124.0** |
> | $\lambda=0.001$  | 54.3     | **86.1** | 119.5     | 153.5     | 43.1     | 60.0     | **74.2** | **86.1** | **41.4** | **66.8** | 95.2     | 125.3     |
> | $\lambda=0.0002$ | **54.2** | 86.5     | 120.2     | 154.1     | **43.0** | **59.9** | 74.2     | 86.2     | 41.6     | 67.6     | 96.2     | 126.1     |
>
> ---
>
> Table 3: Dataset comparison. We compare our dataset with additional multi-person motion datasets employed by previous works for multi-person motion prediction.
>
> | Dataset                     | 2D/3D | Sequences | Frames | Duration (min) | No. of people | Interaction |
> | --------------------------- | ----- | --------- | ------ | -------------- | ------------- | ----------- |
> | KTH Multiview Football [R8] | 2D&3D | 5         | 6.9k   | 3.9            | 2-3           | None        |
> | Haggling [R13]              | 3D    | 34        | -      | 55.6           | 3             | Medium      |
> | UMPM (Interactive) [R1]     | 3D    | 8         | 88k    | 29.6           | 2-4           | Medium      |
> | NTU-RGB+D (SoMoF) [47]      | 3D    | 739       | -      | 28.2           | 2             | Medium      |
> | Ours                        | 3D    | 339       | 60k    | 40.3           | 5             | Strategic   |

---

### Author Response · Authors · 2023-08-14
**Response to Reviewer EKS7 (Repost)**

Thank you for taking the time to provide your feedback. However, we noticed that the specific details of your review were not included. To better understand your viewpoint and address any potential issues, it would be immensely helpful if you could provide more detailed feedback.

---

### Author Response · Authors · 2023-08-14
**Response to Reviewer X2WW (Repost)**

We sincerely appreciate the reviewer's constructive and positive feedback. We are grateful that you recognize our main contributions and find it 'well-written with a clear and well-motivated introduction'. In the following, we aim to address your questions and concerns:

**Q1: Provide more insight on the difference between BC, GAIL and GAN-based supervised learning.**

A1: Wang _et al._ [57] initially introduces an imitation learning formulation for single-person human motion prediction with BC and GAIL compared to end-to-end supervised training. Our work extends conceptually from [24, 57] as it adapts BC and GAIL for a multi-agent imitation learning context. Our approach is distinct from GAN-based supervised learning in two primary ways:

- The GAIL objective focuses on making the state-action pairs produced by the joint policies of agents and that of experts indistinguishable. This is in contrast to the GAN framework (_e.g._, [59]), where the objective is to make the predicted future motion indistinguishable from the real ones.

- Our MARL formulation enables a more principled integration with the cognitive hierarchy theory, while existing works overlook the cognitive aspects of human social action planning. By modeling the motion prediction process with chained policy networks, we derive the BC and GAIL regularization for each level accordingly, and propose to share parameters for policy networks $\phi_{(1)} \dots \phi_{(K)}$, both of which lead to performance improvements as shown in Table 3 of the submitted manuscript.

**Q2: It would be beneficial to provide additional comparisons on existing benchmarks.**

A2: Following your suggestion, we train and test our method on the CMU-Mocap (UMPM) dataset [40]. Please refer to Table 1 of the rebuttal PDF for the results. Our approach outperforms the previous methods [32, R2, 59] and is comparable with concurrent work [40]. Notably, our method exhibits particular strength in predicting long-term root trajectories and global motions.

**Q3: Other multi-person datasets in sports also exist, such as KTH Multiview Football datasets.**

A3: Please refer to General Response Q1 for the discussions. We will make it more clear in the revision.

**Q4: Why was the best performance achieved when K=3?**

A4: Our cognition-inspired framework aims to learn joint policies from real-world expert demonstrations and to discover the explainable cognitive hierarchies. The finding that $K=3$ provides the best fit for the real-world dataset may suggest that this particular depth of strategic reasoning most aptly encapsulates the human decision-making process as captured in our collected data. As we increase the number of hierarchical levels ($K>3$), we continue to observe the cognitive hierarchy visualizations among different levels, though there is a slight increase in the mean prediction error.


**Q5: Could include the visuals in meshes (with ball movement if possible).**

A5: Thanks for your suggestion!

  - **Body Mesh**: As discussed in General Response Q2, we use 3D poses in the experiments. While the 3D poses can certainly be fitted to a parametric body mesh (such as SMPL, as shown in Figure 1) for a more intuitive visualization, this process may introduce additional errors. To avoid these potential complications, we chose to present the 3D skeletons as they are -- the direct output of the model, free from potential distortions introduced by fitting. This approach makes it easier to identify potential issues. However, in response to your suggestion, we will include the mesh fitting and visualization scripts in our code release.

  - **Ball movement**: We implemented a 3D ball trajectory estimator using an object detector and multi-view triangulation. However, we chose not to include the ball information in model training for two reasons: 1) to ensure a fair comparison with previous methodologies, and 2) due to the fact that the ball moves significantly faster than humans, which sometimes result in inaccurate estimations caused by motion blur and detection failures. We are committed to improving our ball detection algorithms, and in response to your suggestion, we will include the 3D ball trajectory estimator and visualization tools in our code release.

---

### Author Response · Authors · 2023-08-14
**Response to Reviewer 8iac (Repost)**

We sincerely appreciate the reviewer’s constructive and positive feedback. We are happy that you find our paper 'well-written', our proposed dataset 'a welcome addition', and our experiments 'very detailed'. In the following, we aim to address your questions and concerns:

**Q1: Lack of survey of related work with respect to the dataset.**

A1: Thank you very much for the suggestion! We provide a discussion on the multi-agent trajectory prediction datasets in General Response Q1. We will cite and discuss these works in Section 2.1 in the next revision following your suggestion.

**Q2: Lack of survey of related work with respect to the methods.**

A2: Thank you very much for the suggestion! We referred to the relevant recent work in the field of multi-agent human trajectory prediction in Line 20-26. We will cite and discuss more research works in this area in Section 2.2 in the next revision following your suggestion. By the way, is part of the review references ([4]) missing?

---

### Author Response · Authors · 2023-08-14
**[1/2] Response to Reviewer Nb5i (Repost)**

We sincerely appreciate the reviewer’s constructive and detailed feedback. We are grateful that you find our paper 'well-written', our proposed 'well-motivated', and our dataset 'highly relevant in the under-explored area'. In the following, we aim to address your concerns:

**Q1: 1s is too short.**

A1: We appreciate the reviewer's insightful suggestions. We understand the importance of longer-term predictions in understanding complex social interactions and the dynamics of the game. However, we would like to provide the following points to clarify our choice of a 1-second forecast horizon:

- Basketball is characterized by its swift pace, whereby a significant amount of action can occur within a 1-second timeframe. In full-court matches, possession often concludes within few seconds. Our research scenario is even more intense and compact compared to conventional basketball games, and it operates at a faster pace.

- Our video demonstration reveals that the 1-second future doesn't merely continue past motions. Instead, it involves defensive switches and sudden changes in moving directions. As a matter of fact, our level-1 policy is primarily motion continuation, but our policy network learns complex strategic behaviors at higher levels through the proposed cognitive hierarchy design.

- We opted not to use a longer prediction horizon primarily because future motions heavily depend on the uncertain outcomes of passes. For instance, if player A is passing to player B, the subsequent motions will be greatly influenced by whether the ball is successfully received by B or stolen midway. Predicting such outcomes based on current information poses a considerable challenge for any model.

We plan to extend our datasets to include regular basketball games with more information and explore longer prediction lengths for future research.


**Q2: Foot sliding.**

A2: Foot sliding might be partly due to our model not explicitly regularizing foot contact velocities as previous works have done [R17, R18, R19]. Our research primarily targets predicting the strategic action decisions rather than the quality of the motion itself. Hence, we did not initially design our model to address these detailed aspects of the motion quality. Based on the reviewer's suggestion, we plan to explore training techniques such as foot contact loss and data augmentation to enhance the visual quality of the motion predictions. We believe these techniques could potentially alleviate the foot sliding issue.


**Q3: Little global translation?**

A3: The no-dribbling rule in this practice enforces the players to make more strategic decisions. While dribbling is prohibited for offensive players, it doesn't necessarily result in a significant decrease in the overall range of motion. It's important to note that defensive players are not under the same restriction. They are required to continually move back and forth to maintain adequate defense coverage. This requirement leads to substantial global translation, as evidenced by the video we have provided. Therefore, while the scope of certain specific motions is reduced, the overall diversity and dynamism of human motions are preserved, and the challenge of predicting those motions remains high. The large root error (especially for the 'frozen' baseline) in Table 2 of the submission also demonstrates this idea.

**Q4: Body joint angles.**

A4: As discussed in General Response Q2, our dataset uses 3D pose representation. Previous work MRT [59] converts the CMU-Mocap dataset to a format of 15-joint 3D poses. We strictly followed its definition and perform data analysis on the aligned skeletal structure. We directly computed the angle of intersection formed by the hinged limbs since we do not have the limb twists of the 3D pose representations.

**Q5: Replicate the MRT experiments.**

A5: Following your suggestion, we train and test our method on the CMU-Mocap (UMPM) dataset [40]. Please refer to Table 1 of the rebuttal PDF for the results. Notably, our method exhibits particular strength in predicting long-term root trajectories and global motions.

**Q6: Details on HRI.**

A6: When training and evaluating HRI, we pass in normalized motions following the original settings. Therefore, it achieves respectable short-term motion prediction accuracy in Table 2, as supposed by the reviewer.

**Q7: CHT overview.**

A7: We provide an intuitive explanation of CHT and level-k reasoning in Line 51-55 and Line 195-197. We will add a short overview to Section 2.3 following your suggestion.

**Q8: Haggling dataset.**

A8: Please refer to General Response Q1. We will cite and compare with it following your suggestion.

**Q9: Intention as intermediate signal.**

A9: Thank you for the information, we will add them to Section 2.

---

> ### Author Response · Authors · 2023-08-14
> **[2/2] Response to Reviewer Nb5i (Repost)**
>
> **Q10: Evaluating the combined motion.**
>
> A10: The GAIL global discriminator $D$ is developed on a similar idea, which penalizes global state-action pairs that are unlikely to occur. For a quantitative evaluation of the combined motion of all persons, incorporating a user study could be beneficial. We appreciate this suggestion and plan to include such a study in our future work.
>
> **Q11: PoseTrack version.**
>
> A11: For PoseTrack, we compare with the version provided by SoMoF [21], which may cause confusion in the number of sequences given different definitions. We apologize for the confusion. We will change the dataset name to 'PoseTrack (SoMoF)' and revise the sequence numbers in the next version.
>
> **Q12: Video encoding error.**
>
> A12: Thank you for bringing this issue to our attention. We have thoroughly reviewed the supplementary video and were unable to replicate the issue. Could you please provide us with your OS and QuickTime Player version? We will ensure that our video encoding is widely compatible in future releases.
>
> **Q13: Color choice.**
>
> A13: Thanks for your suggestion! We will change the visualization colors accordingly.

---

### Author Response · Authors · 2023-08-14
**[1/2] Response to Reviewer yiis (Repost)**

We sincerely appreciate the reviewer's constructive and positive feedback. We are grateful that you are interested in our 'method of technological innovation' with 'good motivation'. In the following, we seek to address your concerns:

**Q1: Compare with more datasets and joints.**

A1: We primarily compare with CMU-Mocap [1] as it is the main dataset used by previous works in this area. We will add more dataset comparisons with other datasets and joints following your suggestion.

**Q2: UMPM and InterHuman dataset.**

A2: Please refer to General Response Q1 for comparisons. We will cite and compare with the datasets following your suggestion.

**Q3: Overclaim data size.**

A3: Thanks for your valuable feedback on our dataset size claim. Nonetheless, it is important to note that the dataset is larger than any previously used in the field of multi-human motion prediction. Our dataset provides 3D skeleton ground truths with intense and strategic human interactions with professional athletes. From such a perspective, we have greatly enlarged the available dataset in this research field for academic uses. We understand how the term "large-scale" might be misconstrued given the temporal length of the dataset. Therefore, in our revised manuscript, we will carefully modify the wording to more accurately represent the size and value of our dataset.


**Q4：Why cannot be used for interactive tasks.**

A4: While it's true that these datasets can be used for motion prediction and generation tasks, their utility for tasks involving interaction - a key focus of our research - is limited. This is due to the relatively low level of interactiveness, as we mentioned in Line 30-32. For instance, the individuals in these datasets are often recorded while moving casually or interacting randomly with others, rather than participating in strategic or purposeful social interactions.

**Q5：NTU-RGB+D.**

A5: NTU-RGB+D contains some action classes with two interacting humans. We provide a comparison in Table 3 of the rebuttal PDF. We will add the comparisons to Table 1 in the revision.

**Q6:  Dataser release.**

A6: The entire dataset, as well as our code and models, will be publicly released.

**Q7: Ablation of $\lambda$s.**

A7: We provide an ablation study on $\lambda$ in Table 2 of the rebuttal PDF.

**Q8: Discuss motion generation.**
A8: Thanks for your suggestion, and our dataset can also be potentially used for motion generation. We will cite and discuss HumanMAC and other related works on motion generation in _Introduction_ and _Related Works_ sections of our revised manuscript.

**Q9: GAIL effects.**

A9: Thanks for your insightful comment. In our manuscript, we have briefly addressed this observation in Line 275-276. Training without GAIL tends to increase the errors in long-term trajectory predictions, while it marginally reduces the errors in local pose prediction. Interestingly, we notice that models trained without GAIL tend to generate motions with smaller magnitudes, often resembling a "freezing" effect. This results in visually less appealing and less realistic animations. Moreover, an important aspect to note is the impact of GAIL on the emergence of cognitive hierarchies within the model. In our experiments, the absence of GAIL leads to a lack of discernible cognitive hierarchy.

**Q10: Verify how the (1)scale, (2)diversity, and (3)interaction benefit the task.**

A10: Thank you for bringing up the important point about the benefits of the scale, diversity, and interaction of our dataset for the social motion prediction task. We are pleased to clarify these aspects:

- Scale: Larger datasets are generally considered beneficial for research tasks as they provide a more extensive base for training models, improving their ability to generalize to unseen data. The advantageous size of our model makes it a valuable resource of the task for the research community.

- Diversity: As shown in Figure 4 and discussed in Line 235-242, our dataset exhibits a high degree of diversity. The results demonstrate that the proposed dataset serves as a more convincing benchmark for social motion prediction and introduces more challenges to the field.

- Interaction: As evidenced by Table 2 and Line 245-249, while the HRI [32] model performs exceptionally well on short-term motion prediction, the global context becomes crucial for long-term motion prediction. Our dataset, with its highly interactive nature, offers strong cues for motion prediction with a global context. These interactions are vital to understanding and predicting complex social behaviors.

We appreciate your feedback and will revise our manuscript to articulate these points more clearly.


**Q11: Annotation.**

A11: Please refer to General Response Q2. We will make it more clear in the revision.

---

> ### Author Response · Authors · 2023-08-14
> **[2/2] Response to Reviewer yiis (Repost)**
>
> **Q12: Motivation for the dataset.**
>
> A12: Thank you for your insightful question.
>
> - Our primary motivation in focusing on intense, strategic interactions stems from a desire to more accurately capture the complexity inherent in real-world human interactions. These interactions often involve strategic decision-making processes and can be social, competitive, or cooperative in nature. Most existing datasets do not adequately capture these complexities, which is a gap that our dataset aims to fill.
> - By incorporating such interactions, we enable the model to learn beyond simple human motion patterns. Our dataset encourages a deeper understanding of human actions, including aspects such as intentions and the decision-making process. Ultimately, our goal is to develop models that not only understand human behaviors but can also interact meaningfully with humans.
> - As for how these properties benefit our task, we refer you to discussions in A10 (Table 4, Figure 2, Line 235-249).
>
> We will revise our presentation to better highlight how these properties of our dataset contribute to the task and benefit the broader research community.

---

### Decision · Program_Chairs · 2023-09-21

**Decision:**

Accept (poster)

**Comment:**

The core contribution of this paper is the introduction of a novel dataset focused on 3D poses in the context of sports. This dataset is particularly intriguing as it delves into the prediction of social fine-grained motions. Given the rising demand and interest in such specific and nuanced data, the dataset holds significant value for the community.

The availability of this dataset can potentially pave the way for advancements in the domain, aiding researchers in understanding and predicting intricate motion dynamics in sports scenarios.

We ask the authors to carefully  integrate reviewers' comments and the ethics discussion in the final version.